

# Automated evaluation systems to enhance exam quality and reduce test anxiety

Doaa Mohamed Elbourhamy

Educational Technology and Computer Department, Faculty of Specific Education, Kafrelshiekh University, Kafrelshiekh, Egypt

## ABSTRACT

University examination papers play a crucial role in the institution's quality, impacting the institution's accreditation status. In this context, ensuring the quality of examination papers is paramount. In practice, however, manual assessments are mostly laborious and time-consuming and generally lack consistency. The last decade has seen digital education acquire immense interest in academic discourse, especially when developing intelligent systems for educational assessment. The presented work proposes an automated system that allows text analysis and evaluation of university exam papers by formal and technical criteria. The research was conducted by analyzing 30 exam papers, which will be included in each of the exam papers, which consist of 60 questions each, in total it holds 1,800 questions. Moreover, it also includes research to understand the quality and relationship with students' test anxiety. A total of 50 year one first-year students were taken to measure students' academic stress by a scale. Planning on basic levels and adherence to technical standards were missing in the exam papers. The proposed automated system has improved exam paper quality to a great extent and reduced academic stress among students with an accuracy of 98% in identifying and matching specified criteria.

## INTRODUCTION

Educational success is fundamental to effective learning and teaching, forming part of a dynamic process that involves the continuous integration of information, decision-making, and responsive planning. At the heart of this process are measurement and evaluation, which assess the stages at which knowledge and skills outlined in the curriculum are realized (*Uysal et al., 2022*). These assessments, conducted before, during, and after instruction, are crucial for fostering learning, developing competencies, and shaping future educational strategies (*Shepard, 2019*; *Ekine & Ebubechukwu, 2021*; *Akçay, Tunagür & Karabulut, 2020*).

However, the pressure associated with assessments, particularly during examination periods, often exceeds students' tolerance, leading to increased academic stress and test anxiety. This stress can significantly impact students' performance, mental well-being, and the overall learning environment (*Dan, Xu & Zhang, 2021*; *Pascoe, Hetrick & Parker, 2020*; *Sun, Dunne & Hou, 2013*). Test anxiety, a specific form of academic stress, arises from the

Corresponding author
Doaa Mohamed Elbourhamy,
doaa.elborhami@spe.kfs.edu.eg

fear of underperforming in exams and the potential consequences of failure. It can negatively affect students' academic outcomes, mental health, and social relationships (*Karaman & Watson, 2017*; *Trigueros, Aguilar-Parra & Cangas, 2020*; *Tudor & Spray, 2017*).

The quality of exam papers plays a critical role in determining the level of test anxiety experienced by students. Poorly designed exams that lack cognitive balance, validity, or reliability can exacerbate stress, while well-constructed assessments can help alleviate unnecessary pressure (*Karatay & Dilekçi, 2019*; *Sharma & Gupta, 2018*). Unfortunately, many teachers lack confidence in their evaluation skills, which can lead to flawed assessments and higher levels of stress for students (*Karatay & Dilekçi, 2019*). This gap in expertise undermines effective learning and proper evaluation (*Shultz, Whitney & Zickar, 2020*).

Automated evaluation systems present a promising solution to address these challenges. These systems play a critical role in modern education by streamlining the assessment process, ensuring objectivity, and enhancing the quality of exam papers. These systems differ significantly from traditional AI-based scoring engines, which primarily evaluate students' test results. Instead, automated evaluation systems focus on analyzing text-based exam content, ensuring alignment with formal and technical criteria, and improving question clarity and structure. By identifying and revising ambiguous or redundant questions, these systems enhance fairness and transparency in assessments. They provide consistent, objective evaluations, significantly reducing the risk of human error and ensuring that exams are fair, comprehensive, and aligned with cognitive principles (*Akçay, Tunagür & Karabulut, 2020*). Furthermore, automated systems improve the quality and reliability of assessments, which directly contribute to reducing student test anxiety by fostering trust in the grading process and creating a transparent, equitable evaluation system. Its primary users are faculty members and administrators, who leverage the system's feedback to refine assessment practices and enhance the overall exam design process. For educators, these tools simplify the evaluation process, promote higher standards in exam quality, and address common challenges such as ensuring cognitive balance and reducing ambiguities (*Nguyen & Habók, 2023*; *Mate & Weidenhofer, 2022*; *Uysal et al., 2022*; *Shepard, 2019*). By aiding educators in creating more effective assessments, automated systems contribute to an improved educational environment and support student success.

This study specifically focuses on addressing test anxiety—a significant component of academic stress—within the context of higher education. By examining the impact of poorly designed assessments on test anxiety, this research highlights the critical need for better evaluation practices in university settings.

The primary objective of this study is to develop and test an automated system that evaluates the quality of university exam papers by ensuring they meet formal and technical standards. This system aims to improve exam consistency and fairness, thereby reducing the academic stress caused by inadequate assessments. Ultimately, this study seeks to promote a culture of consistent measurement and evaluation among educators, leading to enhanced educational outcomes in higher education institutions.

This article is organized as follows: "Related Work" highlights relevant work, "Research Methods" explains the research method, "Results and Discussion" describes the results and discussion, "Finding and Conclusion" outlines the findings and conclusion, "Recommends" provides recommendations, and "Limitations and Future Directions" discusses limitations and future directions.

## RELATED WORK

Previous research has explored the areas of exam quality and academic stress extensively but often in isolation (*Nguyen & Habók, 2023*; *Mate & Weidenhofer, 2022*). There remains a notable gap in literature that integrates these two areas, particularly regarding the potential of automated evaluation systems to improve exam quality while simultaneously reducing student stress. This study aims to bridge this gap by introducing a framework that assesses both exam quality and test anxiety through automated systems, offering a novel approach to understanding their interconnectedness.

To situate this research within the broader context, this section provides a comprehensive overview of two critical aspects in higher education: the quality of exam papers and academic stress. These elements significantly influence educational outcomes and are vital to both student success and educators' effectiveness.

### Exam paper quality

The importance of exam quality in education has been well-documented in the literature. Studies, including those summarized in Table 1, investigate key aspects of exam paper evaluation, from question design to the cognitive demands placed on students. Research shows that high-quality exams, characterized by balanced cognitive challenges and clear formulations, can significantly enhance learning outcomes (*Nguyen & Habók, 2023*). However, few studies explicitly connect exam quality with student well-being, particularly with respect to stress and anxiety.

This research highlights the need to expand existing studies by focusing not only on how exam quality affects learning outcomes but also on how it contributes to student stress. By integrating exam evaluation with psychological factors like test anxiety, we can develop a more holistic understanding of how assessment impacts student performance and mental health.

Previous research has established that test quality significantly influences students' cognitive and emotional responses, drawing on cognitive load theory (*Sweller, 1988*) and test anxiety theory (*Sarason, 1984*). Cognitive load theory emphasizes the role of well-structured assessments in reducing unnecessary mental burden, enabling students to focus on problem-solving rather than deciphering poorly written questions. Test anxiety theory further highlights the interplay between unclear assessment criteria and heightened anxiety, underscoring the need for precise and transparent exam design to promote a fair testing environment. These theories provide a foundation for examining how automated systems can enhance exam quality and indirectly alleviate student anxiety by creating fairer and clearer assessments.

**Table 1 An overview of previous studies in the field of exam paper evaluation.** Various facets of exam paper evaluation, from question formulation to cognitive demand. To complement these findings, new studies must be included that explicitly link exam quality to student performance and stress.

| Research | Research aims | Participant | Methods | Results |
|---|---|---|---|---|
| Evaluation of Student Exam Papers with Respect to Textualization Problems. International (*Kuzu, 2016*) | Demonstrate the textualization problems and issues that students suffer from, by using samples from student exam papers. | The exam papers used in this research belong to 3rd-year students of the Faculty of Education, Department of Turkish Language Education. | Planning of the content based on the awareness level of the content, and establishing subject continuity by clarifying the statement subjects | It is concluded that students, including elementary and secondary education students, need to be educated on the subject of developing answer texts for exam questions, within the scope of text development education, as a part of Writing Skills courses. |
| Turkish teachers' assessment situations: A study on exam papers (*Akçay, Tunagür & Karabulut, 2020*) | The exam papers have been examined from various aspects, including the number and type of questions, the language expression, and the distribution of the questions, the cognitive level (according to Bloom's taxonomy), the type of texts used, and the visuals used. | The samples of the study were selected using a convenience sampling method from 17 secondary schools located in the center of Agri Province in Turkey and the exam papers prepared by 36 Turkish teachers who worked at these schools. The tests which were examined included 2,633 questions in 161 exam papers. | The procedure conducted in the content analysis is to gather information in a certain framework and to interpret such information in a way that the readers can understand | It has been found that the teachers mostly prefer to use the multiple-choice question type in the Turkish exam papers; the questions are mostly related to grammar and reading; the teachers do not prefer to ask questions on speaking and listening; and except for spelling errors, no deficiency is detected in the language of the exam papers. In addition, according to Bloom's taxonomy, the questions are mostly at the comprehension level whereas questions related to the analyzing, evaluating, and creating levels that require high-level thinking skills are rarely used. |
| Analysis of the questions in 11th Grade Philosophy Coursebook in terms of higher-order thinking skills (*Erdol, 2020*) | Investigate the questions included in the 11th grade Philosophy Coursebook prepared in 2018 by the Ministry of Education in Turkey in terms of higher-order thinking skills. | The coursebook included a total of 294 questions within five units | The cognitive domain of the RTB was utilized in the present research since the focus of the research was to analyze philosophy questions in terms of higher-order thinking skills which were part of the cognitive domain | Most of the questions were designed for the mid-level of the cognitive domain and the low-level questions were the second most frequently used questions. Only 6.1% of the questions were designed for high levels of the cognitive domain |
| Considerations and Strategies for Effective Online Assessment in Biomedical Sciences (*Mate & Weidenhofer, 2022*). | To explore the challenges and strategies for implementing effective online assessment tools in the context of biomedical education. The study focuses on how digital assessments can improve the quality and fairness of evaluations. | N/A (The study focuses on a methodological review). | The study uses a qualitative review method, evaluating different online assessment strategies and tools in the context of biomedical sciences. | Implementation of automated systems significantly reduces grading errors and enhances perceived fairness, leading to improved student satisfaction and assessment quality. |

| Research | Research aims | Participant | Methods | Results |
|---|---|---|---|---|
| Examining Turkish Course Exam Questions in Terms of Item Writing Criteria and Cognitive Level (*Uysal et al., 2022*). | To analyze Turkish language exam questions based on item writing criteria, cognitive level, and question format. The study aims to identify the strengths and weaknesses in current exam design. | 747 questions from 51 exam papers in Turkish language courses. | Document analysis and maximum variation sampling methods were used to evaluate the exam questions. | The analysis revealed that the majority of questions were focused on understanding and recall, with fewer questions targeting higher-order thinking skills such as analysis and creation. Automated evaluations help educators identify these gaps and improve exam quality. |
| Tools for Assessing Teacher Digital Literacy: A Review (*Nguyen & Habók, 2023*). | To review and analyze the tools available for assessing teacher digital literacy, with a focus on their effectiveness in enhancing exam quality and teacher evaluations. | N/A (The study is a literature review). | The study employs a systematic review approach, analyzing various digital literacy tools and their impact on teacher performance and evaluation practices. | Automated tools for teacher assessment contribute to more consistent and objective evaluations, supporting teachers in designing higher-quality exams aligned with educational standards. |

## Academic stress and test anxiety

Academic stress, particularly test anxiety, is a prevalent issue in higher education, affecting a significant portion of the student population (*Pascoe, Hetrick & Parker, 2020*). Table 2 provides a summary of research exploring the various factors contributing to academic stress, including cognitive load, exam clarity, and question complexity. Studies reveal that poor exam design can exacerbate stress levels, leading to diminished academic performance and negative psychological outcomes (*Trigueros, Aguilar-Parra & Cangas, 2020*).

While much of the literature focuses on academic stress as an isolated issue, there is limited research on how specific exam design elements contribute directly to test anxiety. This study extends current research by exploring the link between exam paper quality and test anxiety, offering insights into how well-designed exams can alleviate stress.

## Automated evaluation systems

The advent of automated evaluation systems in education introduces new possibilities for improving both the consistency and fairness of assessments. Previous research has primarily focused on how these systems enhance grading efficiency and reduce workload for educators (*Nguyen & Habók, 2023*). However, little attention has been given to how these systems might also reduce stress for students by providing objective, bias-free evaluations.

Automated systems not only minimize human error in grading but also ensure uniformity in assessments, which can help reduce the anxiety that students often experience from perceived grading inconsistencies (*Pascoe, Hetrick & Parker, 2020*). For educators, these systems free up time by automating routine grading tasks, allowing them to focus on providing more meaningful feedback. This dual impact on both educators and

**Table 2 Relevant studies for academic stress.** The critical research on academic stress, a prevalent issue in higher education.

| Research | Research aims | Participant | Methods | Results |
|---|---|---|---|---|
| Academic Stress in University Students: Systematic Review (*Serveleon Quincho et al., 2021*) | Examine how academic stress develops in university students. | University students. | Systematic review of academic articles from 2018 to 2020. | Excessive academic load identified as a major stressor. |
| The Impact of Stress on Students in Secondary School and Higher Education (*Pascoe, Hetrick & Parker, 2020*) | To systematically review the main sources of academic stress in secondary and higher education, with a focus on high-stakes exams and their impact on student well-being. | Secondary and higher education students (review of multiple studies). | Systematic literature review of previous studies on academic stress, exam quality, and student well-being. | High-stakes exams are a major contributor to academic stress. Enhancing the quality of exams can reduce anxiety and promote better educational outcomes. |
| Who's Stressed? Distributions of Psychological Stress in the United States in Probability Samples from 1983, 2006, and 2009 (*Cohen, Janicki-Deverts & Miller, 2007*). | Assess psychological stress in the US and understand stress distributions across demographic groups and time. | US adult residents from surveys conducted in 1983, 2006, and 2009. | Analysis of national surveys using the Perceived Stress Scale. | Stress higher among women, younger adults, those with lower socioeconomic status, and certain demographics like middle-aged, college-educated White men. |
| The Influence of Emotional Intelligence on Resilience, Test Anxiety, Academic Stress, and the Mediterranean Diet: A Study with University Students (*Labrague & McEnroe-Petitte, 2016*). | Analyze the influence of emotional intelligence on resilience, academic stress, exam anxiety, and eating habits related to the Mediterranean diet. | 733 male and 614 female students, aged 19–27, from the University of Almeria. | Structural equation modeling to explain causal relationships between variables. | Emotional intelligence positively predicted resilience; resilience negatively predicted test anxiety and academic stress; test anxiety and academic stress negatively predicted adherence to the Mediterranean diet. |
| Assessment of Stress Biomarkers in Students (*Murphy, Denis & Ward, 2017*). | Examine salivary biomarkers as indicators of academic stress in students. | Various student groups including undergraduate, medical, and dental students. | Measurement of salivary cortisol and other biomarkers in response to academic stress. | Identified salivary biomarkers indicative of stress responses in academic settings. |
| Academic Stress, Parental Pressure, and Test Anxiety Among Chinese High School Students (*Sun, Dunne & Hou, 2013*). | To examine the impact of exam quality and parental expectations on academic stress and test anxiety among Chinese high school students. | High school students in China. | Mixed-method approach, including surveys to measure academic stress and interviews to explore students' perceptions of exam quality. | Poorly structured exams and high parental expectations contribute significantly to test anxiety and academic stress, suggesting a need for more student-centric exam design. |

students highlights the potential for these systems to be more than just efficiency tools—they are integral to creating a more supportive learning environment that addresses both academic and psychological needs.

## Automated evaluation systems in education

The growing complexity of educational assessment, coupled with the increased number of students in higher education, has led to a rising demand for automated evaluation systems. These systems provide several advantages, including consistency in grading, reduction of human errors, and efficiency in managing large-scale evaluations (*Nguyen & Habók, 2023*). Automated systems have been shown to significantly reduce the workload for evaluators, particularly when handling repetitive tasks like grading multiple-choice or true/false

questions, allowing them to focus more on creative and critical-thinking assessments (*Mate & Weidenhofer, 2022*).

Studies have demonstrated the effectiveness of machine learning models, such as Naive Bayes and SVM, in automating the evaluation process. For instance, *Nguyen & Habók (2023)* achieved 90.2% accuracy using Naive Bayes, while *Mate & Weidenhofer (2022)* achieved 88.5% with an SVM approach. However, these models primarily focus on evaluating multiple-choice questions, leaving a gap in evaluating more complex, open-ended responses.

By addressing this research gap, our proposed automated evaluation system introduces an integrated framework that not only handles technical and formal criteria in exam papers but also ensures a fair and consistent evaluation of diverse question formats. Unlike prior approaches that focus solely on accuracy, our system also emphasizes reducing evaluator stress and improving overall grading efficiency (*Elbourhamy, 2024*).

## Research gaps and the need for an integrated approach

Despite extensive literature on exam quality and academic stress, there remains a lack of research that integrates these areas, particularly through the use of automated systems. This study addresses this gap by proposing a framework that evaluates both exam quality and test anxiety concurrently. By developing an automated system that evaluates formal and technical aspects of exam papers and correlates them with student stress levels, this research offers a comprehensive and objective method of assessment.

## Significance of the proposed research

The proposed research introduces a novel approach to exam paper evaluation by combining automated analysis with psychological metrics, such as test anxiety. By linking exam quality to student well-being, this study aims to revolutionize how assessments are conducted in higher education. For educators, the system offers actionable insights into how exam design influences student stress levels, enabling improvements in exam construction. For students, the automated system ensures high-quality, fair exams that are not only cognitively challenging but also mindful of their psychological well-being.

In conclusion, this study emphasizes the critical role that exam structure and presentation play in student success and well-being. By automating the evaluation process and correlating exam quality with academic stress, this research seeks to inform and enhance both educator training and student outcomes, ultimately aiming to reduce test anxiety and improve performance in higher education.

The key research question guiding this study is: "*How effective is the proposed automated system in evaluating exam papers, and what is its impact on students' test anxiety in higher education?*"

Therefore, this study proposes an innovative automated model for evaluating exam papers, utilizing specific formal and technical evaluation criteria, such as university, faculty, course, question headers, repeated questions, and repeated alternatives, *etc*. The goal is to foster a culture of rigorous measurement and evaluation among educators and students, supporting instructors in designing high-quality exam papers that not only meet

formal standards but also reduce students' academic stress. This, in turn, will enhance the efficiency and well-being of educational processes in higher education institutions.

## RESEARCH METHODS

This study employs a rigorous methodology to assess the effectiveness of an automated system for evaluating university exam papers and its impact on students' test anxiety. The method comprises two main parts: the development and deployment of an automated system for evaluating exam papers, and the analysis of the system's effect on academic stress levels among students.

### Data collection

The data for this study was collected from 50 first-year English-language computer science teacher students at the Faculty of Specific Education, Kafrelsheikh University. Data collection occurred over two semesters (2021/2022 academic year), with assessments of academic stress levels taken both before and after the implementation of the automated system. Written informed consent was obtained from all participants, with the option of withdrawal without penalty.

The automated system was applied to 30 exam papers, each containing approximately 60 questions, totaling 1,800 questions. The implementation occurred in two phases: after the first-semester exams and again after the second-semester exams, focusing exclusively on English-written exams.

The evaluation criteria used in this study were developed based on an extensive review of the literature on exam quality and assessment standards (*Akçay, Tunagür & Karabulut, 2020*; *Karatay & Dilekçi, 2019*; *Shepard, 2019*), and in consultation with experts in measurement and evaluation. These criteria, officially adopted by the Faculty of Specific Education at Kafrelsheikh University, focus on formal and technical aspects, rather than cognitive levels.

-Formal criteria include the presence of essential information such as university and course names, exam date, time allocation, and total exam score, ensuring compliance with institutional standards (*Nguyen & Habók, 2023*).

-Technical criteria cover the structure and clarity of the exam content, such as clear question headers, non-redundant questions, and the avoidance of ambiguous options like "all of the above" or "none of the above" (*Mate & Weidenhofer, 2022*; *Shultz, Whitney & Zickar, 2020*).

By implementing these criteria, the automated evaluation system aims to provide a comprehensive review of exam papers, ensuring adherence to institutional standards and reducing inconsistencies that could contribute to student stress. Table 3 summarizes the formal and technical criteria used in the evaluation.

### Development of the automated evaluation system

The proposed automated system was developed to assist instructors in refining exam papers and improving their quality. Figure 1 provides an overview of the system's architecture, detailing its key components and the process flow.

**Table 3 Formal and technical criteria.** Each data demonstrates both formal and technical criteria, providing detailed descriptions of each.

| Criteria | | | Description |
|---|---|---|---|
| Formal criteria | 1 | University | Existing university name |
| | 2 | Faculty | Existing faculty name |
| | 3 | Course | Existing course name |
| | 4 | Program | Existing program name |
| | 5 | Exam date | Existing exam date |
| | 6 | Level | Existing level |
| | 7 | Semester | Existing semester |
| | 8 | Department | Existing department name |
| | 9 | Exam time | Existing exam time |
| | 10 | Academic year | Existing academic year |
| | 11 | Total exam score | Existing total exam score |
| Technical criteria | 12 | Question headers | Existing question headers |
| | 13 | Repeated questions | Existing repeated questions |
| | 14 | Repeated alternatives | Existing repeated alternatives |
| | 15 | "All of the above" | Existing "all of the above" |
| | 16 | "Only" | Existing "only" |
| | 17 | Number of questions | How many questions are in the exam paper? |
| | 18 | Types of questions in the exam | Show types of questions in the exam |
| | 19 | The number of questions corresponds to the exam time | Does the number of questions correspond to exam time? |

**Key features of the automated system:**

*1. Upload exam paper:* Instructors upload exam papers as PDF files.

*2. PDF to text conversion:* The system uses the PyPDF2 library to extract text from the PDF, which is then concatenated into a single string for further processing.

$$Text = \sum_{i=1}^{n} page_i. \tag{1}$$

- Text: The concatenated text from the entire PDF.
- n: The number of pages in the PDF.
- $page_i$: The text extracted from the i-th page of the PDF.

*3. Text preprocessing:* The text is preprocessed by converting it to lowercase, removing special characters, and correcting spelling errors.

$$Cleaned\ Text = Lowercase(RemoveSpecialChars(Text)). \tag{2}$$

- *Cleaned Text*: The preprocessed text.
- Lowercase(x): Converts text x to lowercase.
- RemoveSpecialChars(x): Removes special characters from text x.

*4. Tokenization and Stemming:* The nltk library's 'word_tokenize' function is used to tokenize the text into individual words, while the PorterStemmer

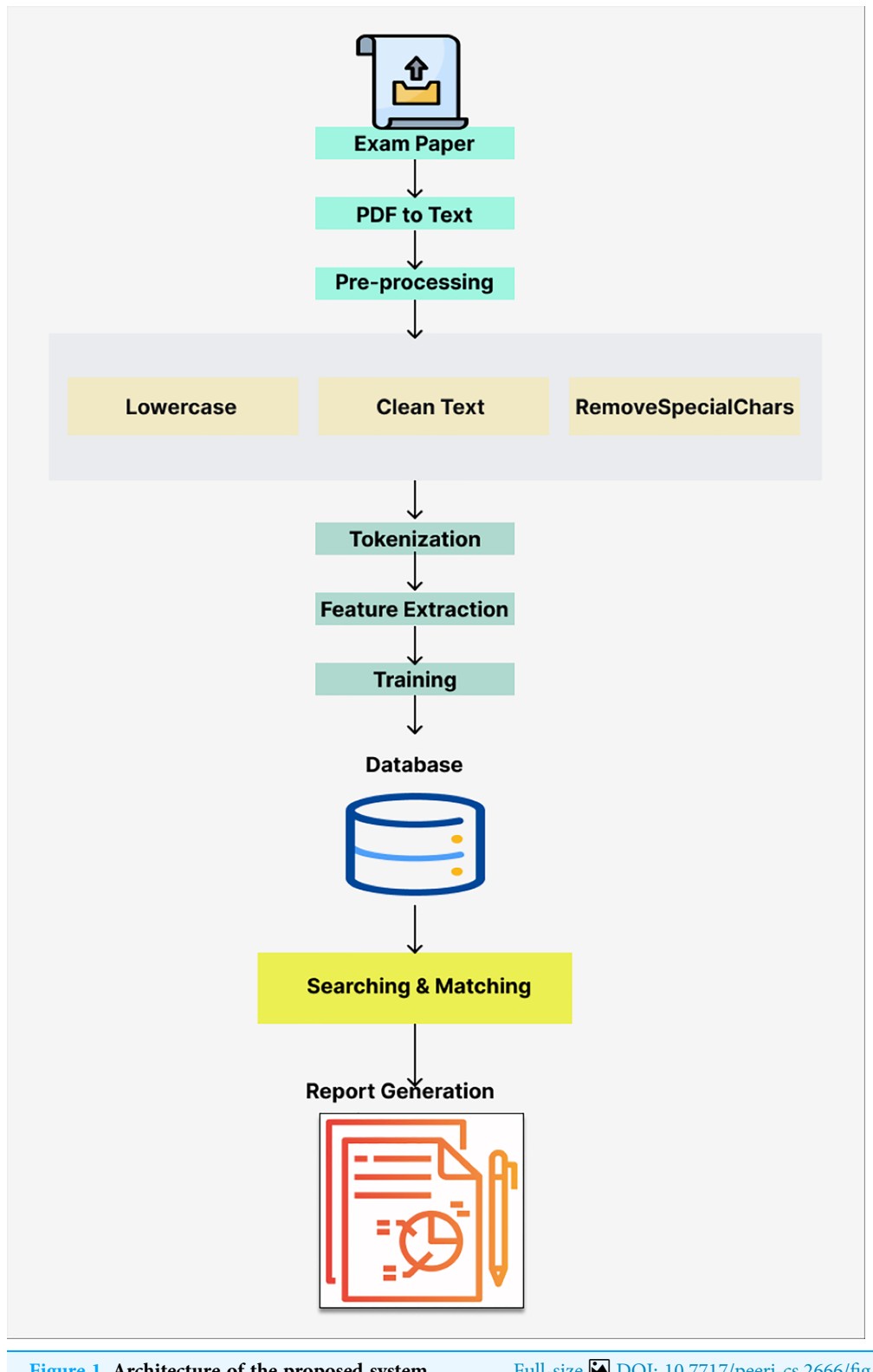

**Figure 1 Architecture of the proposed system.**

reduces words to their root forms (*Vijayarani & Janani, 2016*; *Mohammdi & Elbourhamy, 2021*).

$$\text{Tokens} = \text{word\_tokenize}(\text{Cleaned Text}). \qquad (3)$$

- Tokens: The list of tokenized words.
- word_tokenize(x): Tokenizes the text x into individual words.

**5. Feature extraction:** The TF-IDF vectorizer from the scikit-learn library transforms tokenized text into numerical features suitable for machine learning algorithms.

$$\text{TF} - \text{IDF}(t, d, D) = \text{TF}(t, d) \times \text{IDF}(t, D). \qquad (4)$$

- $\text{TF} - \text{IDF}(t, d, D)$: The TF-IDF score of term t in document d within the document set D.
- $\text{TF}(t, d)$: The term frequency of term t in document d.
- $\text{IDF}(t, D)$: The inverse document frequency of term t in the document set D.

**6. Model training:** A naive Bayes classifier is trained on labeled exam questions to classify formal and technical criteria. The model was trained using labeled data and evaluated on a test set to ensure accuracy.

$$P(c|d) = \frac{(d|c).P(c)}{P(d)}. \qquad (5)$$

- $P(c|d)$: The probability of class c given document d.
- $(d|c)$: The probability of document d given class c.
- $P(c)$: The prior probability of class c.
- $P(d)$: The prior probability of document d.

**7. Criteria database:** The SQLite-based criteria database was developed to store the formal and technical criteria. This dynamic, user-friendly database allows administrators to add or remove criteria as needed, ensuring that the system remains flexible and up-to-date. Figure 2 shows the administrative panel of the database.

**8. Searching and matching:** The system searches for keywords that match the criteria stored in the database. The trained naive Bayes model predicts the matching criteria for new exam papers.

$$\text{Predicted criteria} = \text{Model.predict}(\text{TF} - \text{IDF Features}). \qquad (6)$$

- Predicted criteria: The predicted criteria for the new text data.
- Model.predict(x): The prediction of the model for features x.

**9. Automated question counting:** A Python script automates the counting and classification of multiple-choice and true/false questions using regular expressions. This ensures accurate and efficient detection of question types.

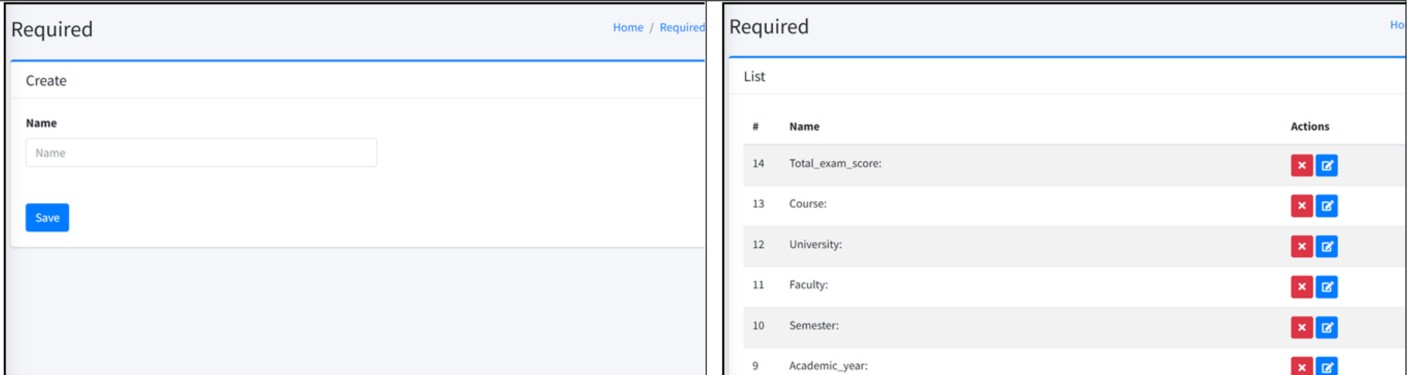

**Figure 2 GUI of administration database panel.** A snapshot of the functionality that allows for comprehensive control over the database contents.

**Libraries used:**

- **PyPDF2:** A library for reading PDF files.
- **re (Regular Expressions):** A library for matching patterns in text.

**Explanation of the process:**

1. **extract_text_from_pdf:** Extracts text from the PDF file.
2. **identify_question_patterns:** Defines regex patterns for identifying multiple-choice, true/false, and short answer questions.
3. **classify_questions:** Uses regex patterns to classify questions and returns lists of multiple-choice, true/false, and short answer questions.
4. **main:** Orchestrates the process of extracting text, classifying questions, and printing the results.

This automated approach ensures accurate and efficient counting of questions in exam papers, facilitating better exam preparation and analysis.

***10. Evaluation of exam time correspondence:*** To evaluate whether the number of questions corresponds to the exam time, we set standard average times for each type of question based on recent educational research:

- **Multiple-choice question:** 1.5 min (*Cognitive Research: Principles and Implications, 2023*; *Evidence Based Education, 2022*).
- **True/False question:** 1 min (*International Journal of STEM Education, 2023*).

For an exam with $N_{MC}$ multiple-choice questions and $N_{TF}$ true/false questions, the expected total time needed is calculated as follows:

**1. Expected time for multiple-choice questions:**

$$\text{Expected Time}_{MC} = N_{MC} \times \text{Average Time per MC Question}. \tag{7}$$

**Parameters:**

- $N_{MC}$: The total count of multiple-choice questions in the exam.
- Average Time per MC Question: The average time allocated to answer each multiple-choice question is 1.5 min.

**2. Expected time for true/false questions:**

$$\text{Expected Time}_{TF} = N_{TF} \times \text{Average Time per TF Question} \tag{8}$$

**Parameters:**

- $N_{TF}$: The total count of true/false questions in the exam.
- Average Time per TF Question: The average time allocated to answer each true/false question, set to 1 min.

**3. Total expected time:**

$$\text{Total expected time} = \text{Expected Time}_{MC} + \text{Expected Time}_{TF}. \tag{9}$$

**Parameters:**

- $\text{Expected Time}_{MC}$: The total expected time to answer all multiple-choice questions.
- $\text{Expected Time}_{TF}$: The total expected time to answer all true/false questions.

Given that the actual allotted exam time ($T_{exam}$) is 120 min, we compare it with the total expected time ($T_{total}$). If ($T_{total}$) ≤ ($T_{exam}$), then the number of questions corresponds well with the exam time, ensuring that students have ample time to complete the exam. Otherwise, adjustments to the number of questions or the allotted time are necessary.

***11. Report generation:*** The system compiles the results into a detailed report that includes both the formal and technical criteria of the exam paper, along with the number and types of questions and the exam's time correspondence. This comprehensive report serves as a tool for instructors to refine and enhance the quality of their exam papers. The report can be conveniently printed or emailed directly to the instructor or the educational institution's Measurement and Evaluation department.

## Evaluation metrics and procedures

The development of the proposed automated system was guided by the goal of creating an effective and practical tool tailored specifically for educational institutions, particularly universities. This system was engineered using Python 3.11, PHP, and the Laravel framework, providing a robust and scalable foundation for evaluating exam papers. The primary aim of the system is to support teachers in evaluating exam papers efficiently and to reduce students' academic stress by ensuring exam fairness and consistency.

### System design and user interface

The user experience was a critical focus during the system's development. The design features a user-friendly interface to simplify the evaluation process. The homepage offers an intuitive "Choose File" option, where instructors can upload exam files in PDF format (Fig. 3). This design choice ensures ease of use, allowing users to initiate the evaluation process without complexity.

The system evaluates the uploaded exam papers in six sequential stages, each focusing on specific formal and technical criteria. These stages were designed to streamline the evaluation process and ensure comprehensive checks of exam papers against key standards.

Stage 1: File upload and initial interface

-Users begin by uploading the exam file in PDF format through the homepage interface. Once the file is selected, users proceed by clicking the "Next | Upload" button, moving to the next phase.

Stage 2: Formal criteria check

-In the second stage, the system scans the exam paper for key formal criteria, such as the presence of terms like 'university' and 'college' (Fig. 4). This is initiated by clicking the "Check Required" button. The system then validates whether the exam paper meets the required institutional standards, providing immediate feedback to users.

Stage 3: Forbidden words identification

-The third stage evaluates the presence of undesired words in the exam paper, such as 'only' and 'all of the above' (Fig. 5). Users initiate this stage by selecting the "Check Forbidden" button. The system scans the document, identifying any forbidden phrases that may cause ambiguity or unfairness in the assessment.

Stage 4: Duplicated questions detection

-In the fourth stage, the system checks for duplicated questions (Fig. 6). By clicking the "Check Duplicated Questions" button, users can ensure that no question is repeated within the exam, maintaining the uniqueness and validity of the assessment.

Stage 5: Duplicated answers in multiple-choice questions

-The fifth stage involves identifying redundant alternatives in multiple-choice questions (Fig. 7). Clicking the "Check Duplicated Answers" button allows the system to verify that no answer choice is repeated across multiple questions, thus ensuring fairness and avoiding confusion.

Stage 6: Question header, number, and time analysis

-In the sixth stage, the system verifies the presence of question headers and evaluates the number of questions relative to the allocated exam time (Fig. 8). This analysis ensures that the exam length and complexity are proportionate to the available time, addressing one of the key concerns related to student stress—time management during exams.

Final Stage: Comprehensive report generation

The final stage involves generating a comprehensive report summarizing the exam's evaluation across all formal and technical criteria (Fig. 9). This report not only indicates whether the exam adheres to institutional standards but also provides an analysis of other

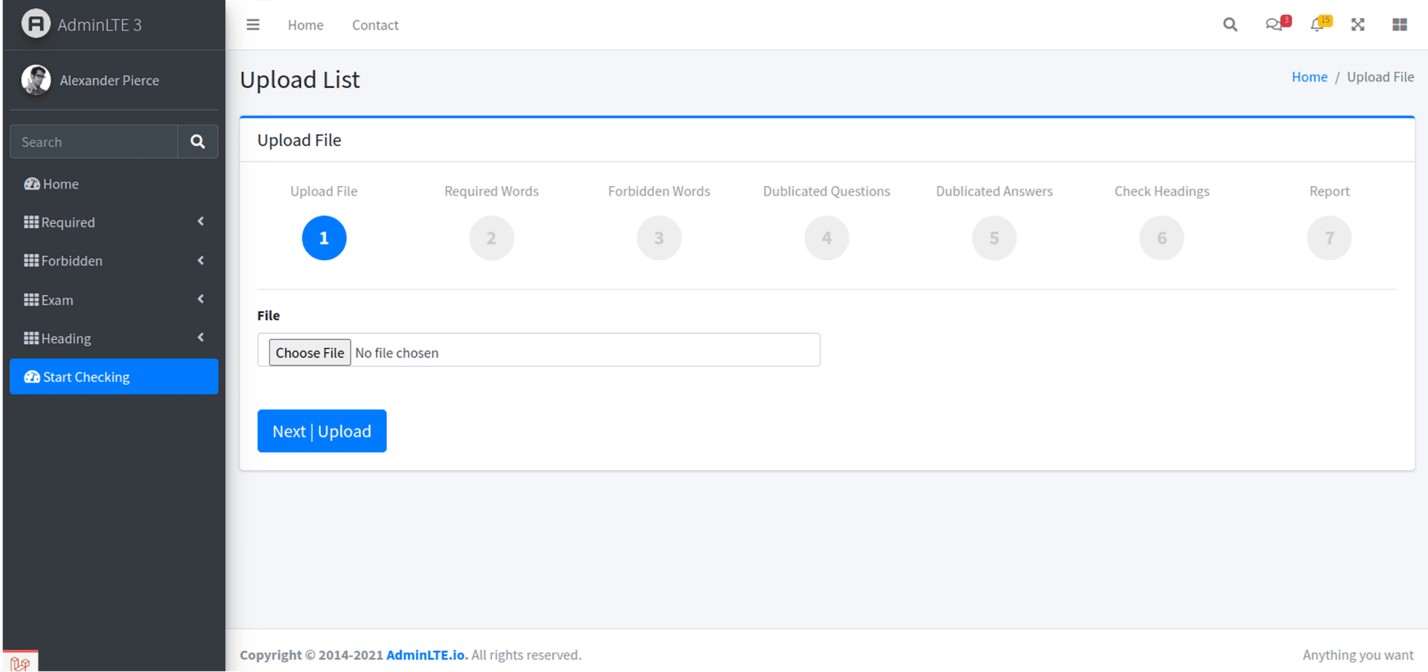

**Figure 3** The homepage of the proposed system's user interface, showcasing the initial exam paper evaluation process stage.

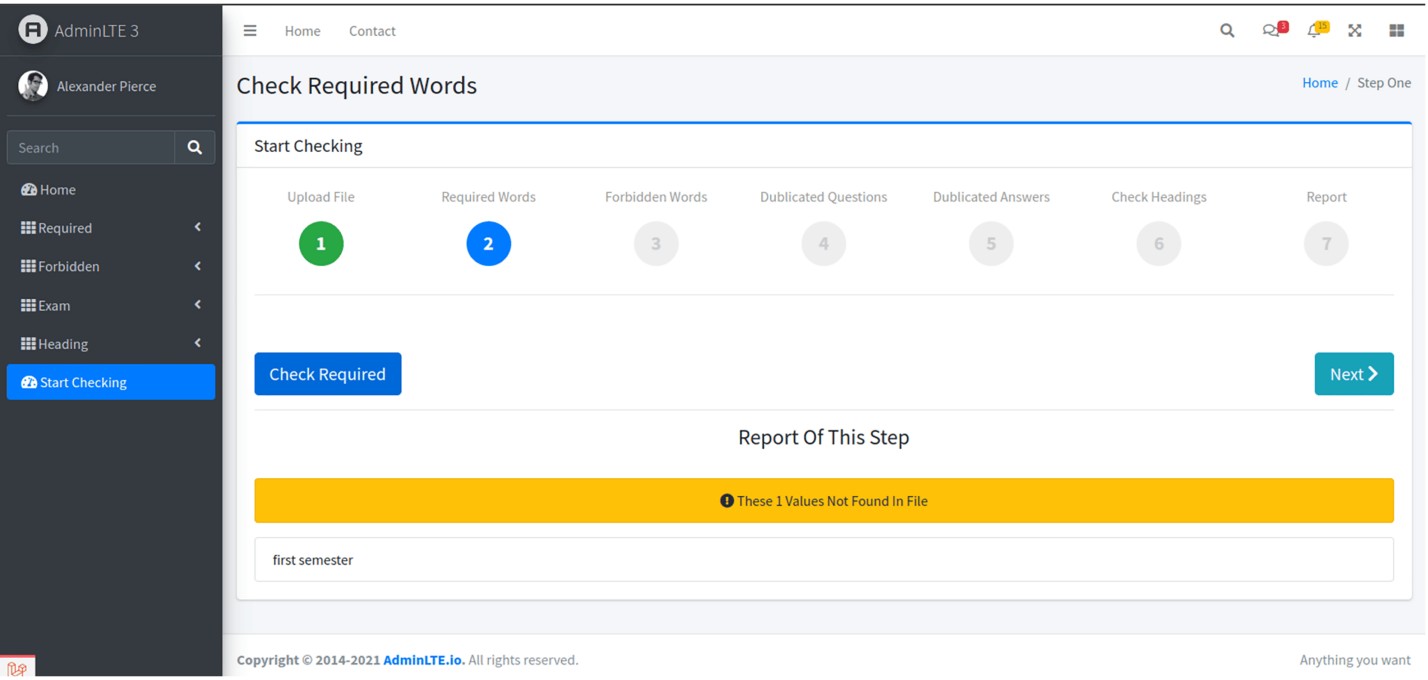

**Figure 4** **The second stage.** This is achieved by clicking on the 'Check Required' button, which triggers the system to validate the presence of these criteria in the exam paper.

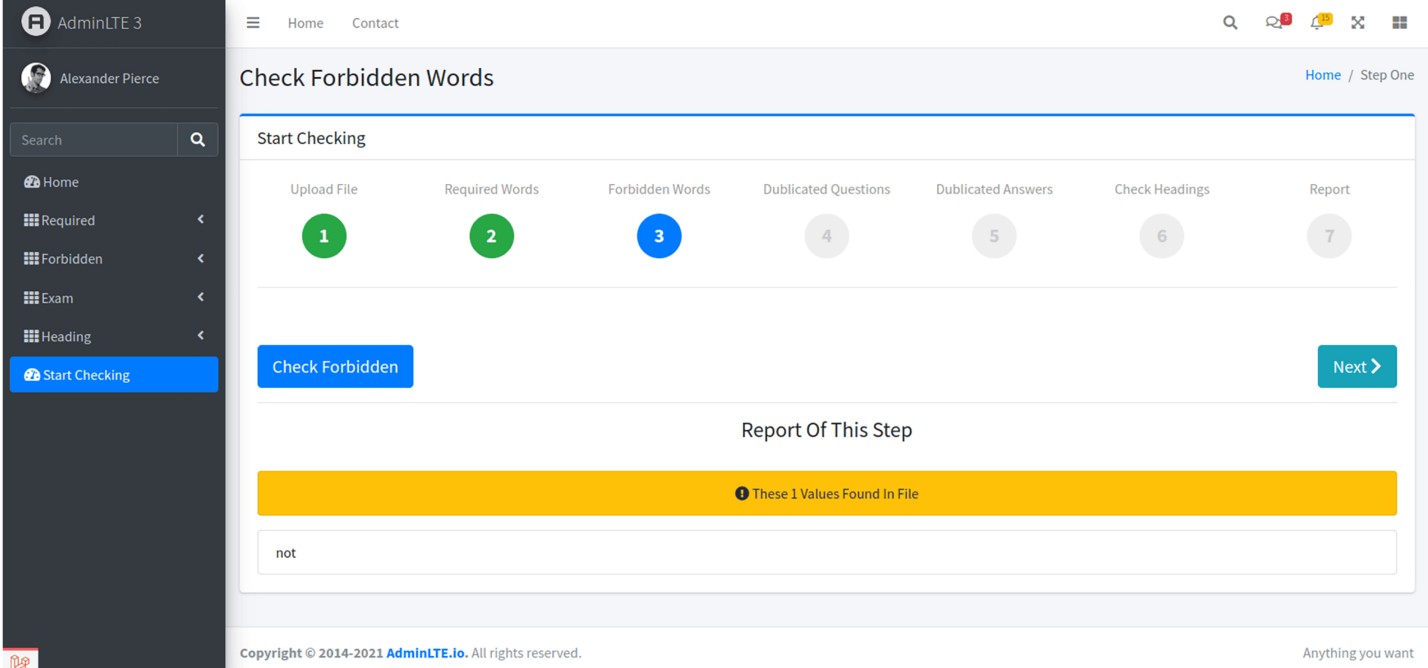

**Figure 5** **The third stage.** The system analyzes the uploaded exam paper to identify undesired words, such as 'only' and 'all of the above.' This is executed by selecting the 'Check Forbidden' button.

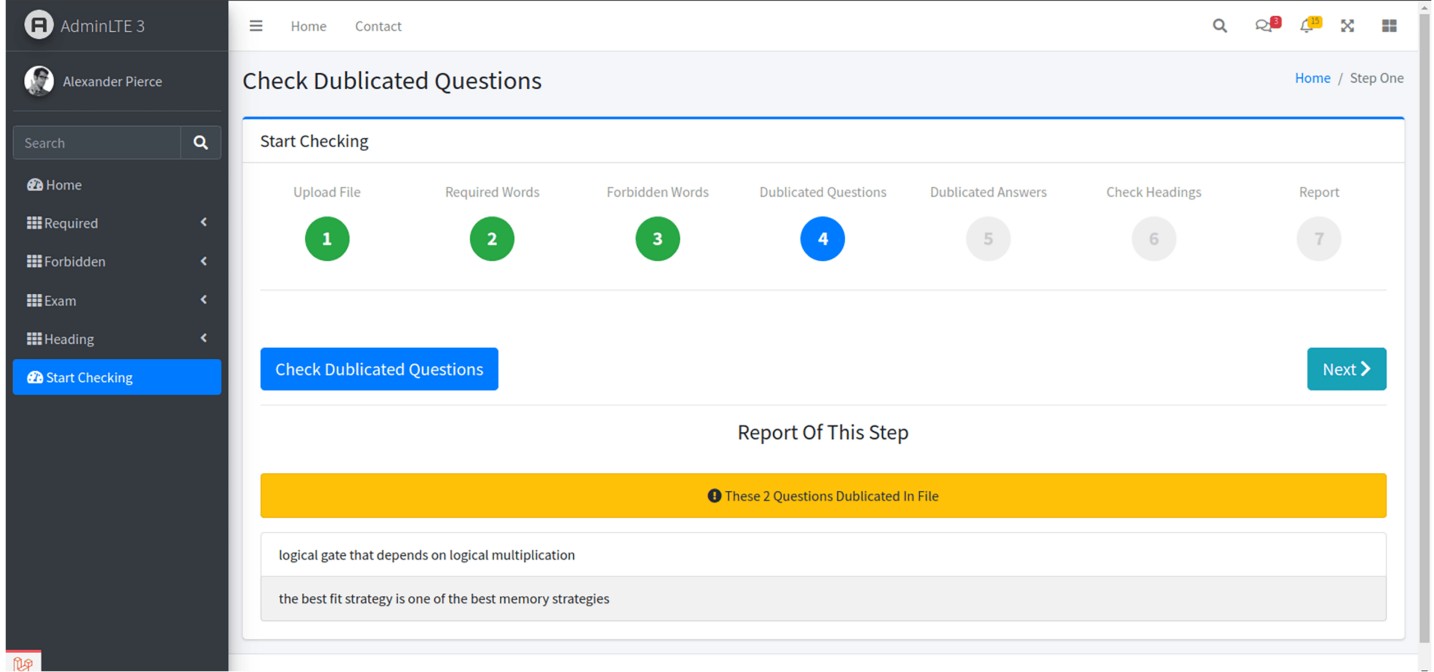

**Figure 6** **The fourth stage.** This step begins by clicking the 'Check Duplicated Questions' button. Upon activation, the system performs a comprehensive search for duplicate questions. The results are visually displayed where two questions are identified as duplicates.

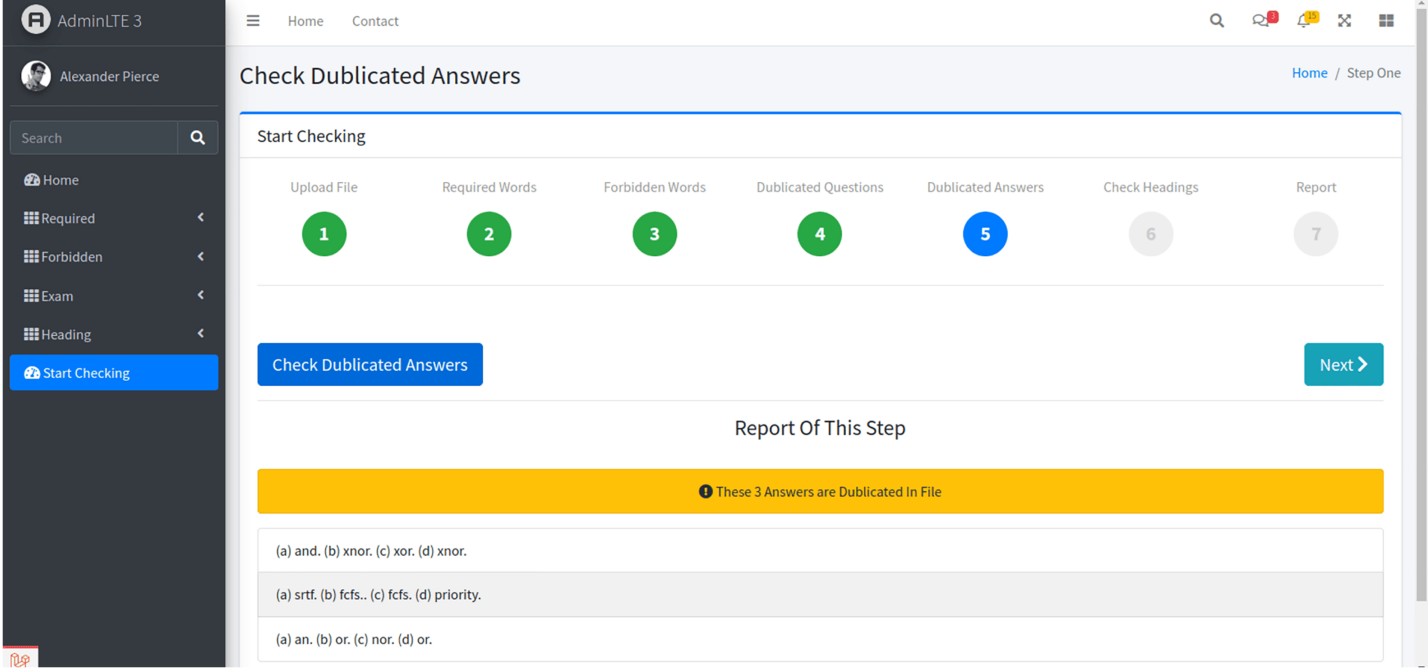

**Figure 7** **The fifth stage.** The system analyzes the exam paper for this specific type of redundancy, and the findings.

factors such as question type distribution and time alignment. The report can be exported as a PDF file and emailed directly to educational institution officials for further review or action.

### System evaluation and real-world application

The evaluation of the system's performance focused on its accuracy, usability, and effectiveness in improving the quality of exam papers. The model training process utilized a dataset composed of exam papers annotated by a team of expert educators. This dataset included 30 exam papers, each containing approximately 60 questions, drawn from various courses in the English-language computer science program. The questions were categorized into multiple-choice and true/false formats. Expert annotations provided labels for both formal criteria (*e.g.*, inclusion of course title, exam date, and total score) and technical criteria (*e.g.*, clarity, redundancy, and alignment with cognitive principles), ensuring the reliability and consistency of the training data.

The naive Bayes classifier was trained using 80% of this annotated dataset, while the remaining 20% served as a test set to evaluate the model's performance. To ensure robustness, tenfold cross-validation was employed during training, fine-tuning hyperparameters, and verifying the classifier's predictive accuracy. Evaluation metrics, including precision, recall, and F1-score, were used to assess the model's reliability in predicting both formal and technical criteria. Feedback from teachers and educational
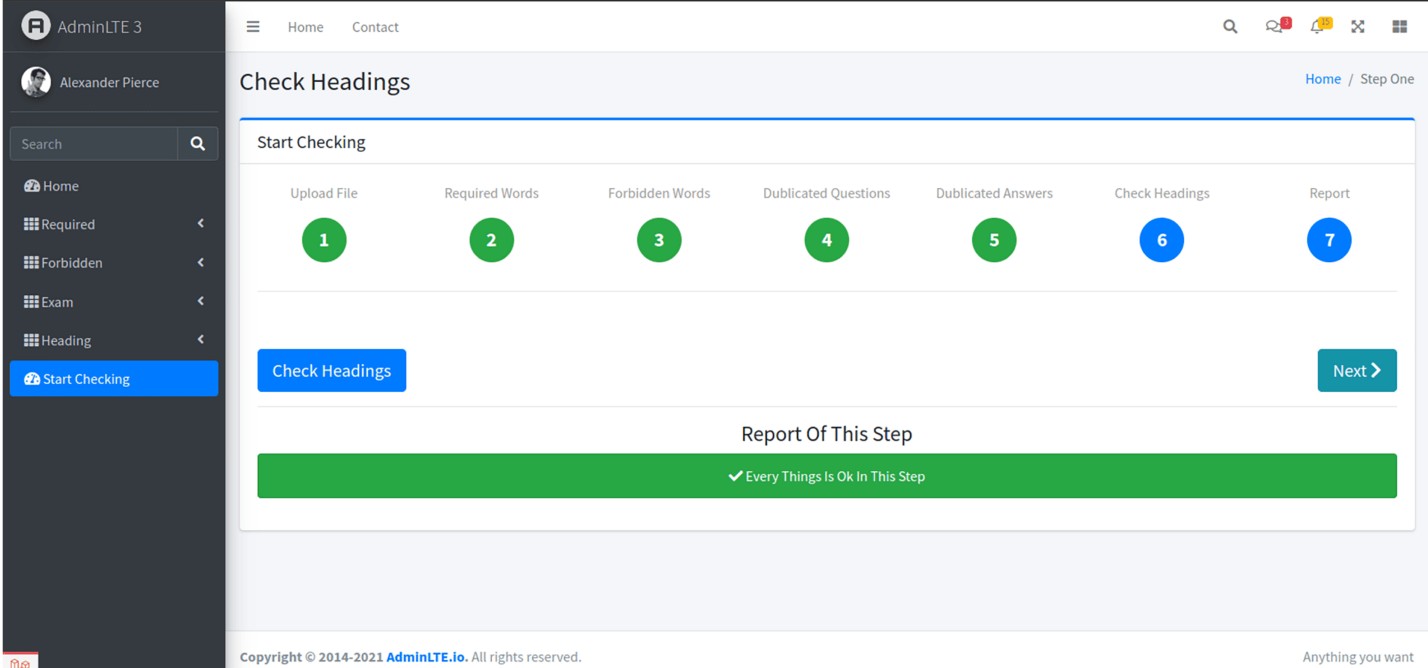

**Figure 8** **The sixth stage.** This stage scrutinizes the exam paper to verify the presence of a question header and to assess the number and type of questions, ensuring that the quantity of questions is proportionate to the allotted exam time.

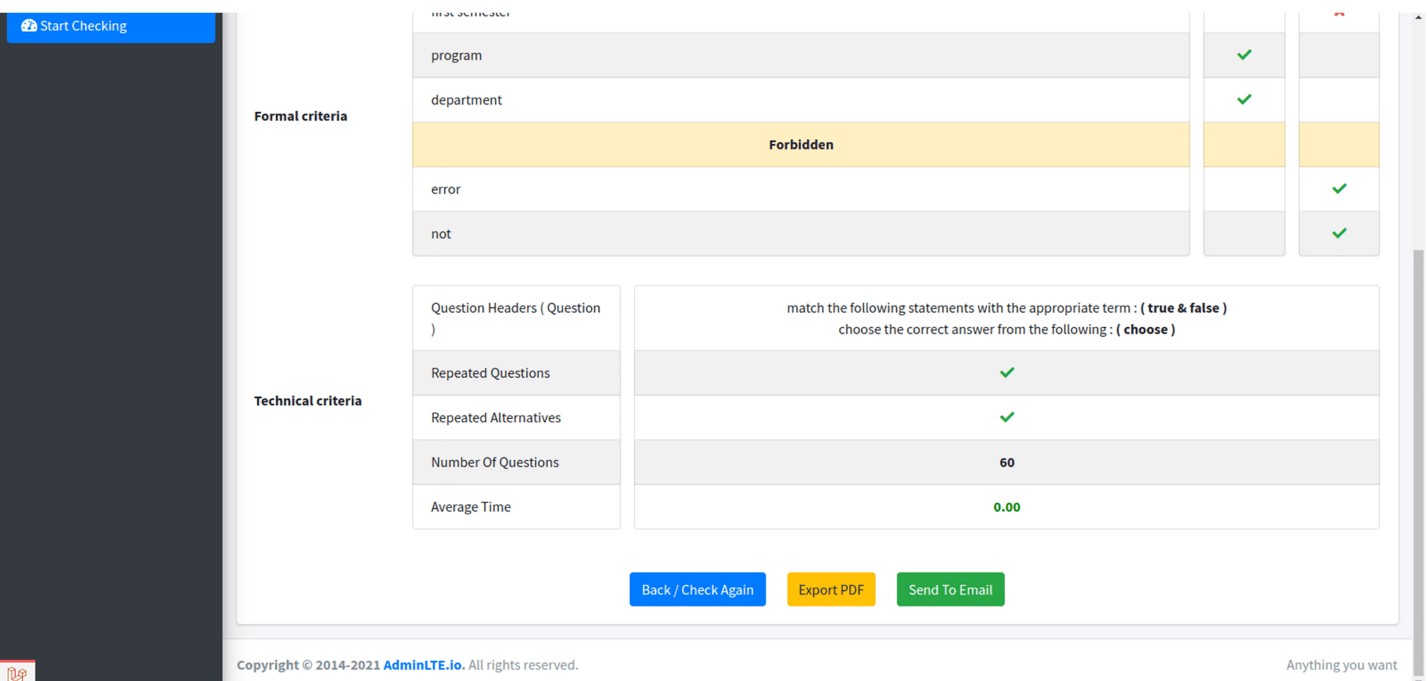

**Figure 9** **The final report for the exam paper.** The generation of a comprehensive report. This report encompasses all the formal and technical criteria assessed in the previous stages for the examined exam paper.

administrators was gathered, and pre- and post-intervention stress levels were analyzed to determine the system's impact on reducing student test anxiety.

Key features of the system, such as automatic checks for formal and technical criteria, ensured fair and consistent assessments. By addressing common sources of student stress, such as ambiguous questions or unrealistic time limits, the system contributed to a significant reduction in test anxiety. Moreover, aligning exam length and complexity with student time management enhanced the educational environment by fostering transparency and trust. In summary, the proposed system revolutionizes the evaluation process by automating key elements of exam paper evaluation, providing educators with a reliable tool to maintain high-quality exam standards while directly contributing to reducing student test anxiety.

## Evaluation of the system

The effectiveness and efficiency of the proposed automated evaluation system were rigorously assessed through accuracy testing and analysis of the system's impact on students' academic stress. This section outlines the evaluation process, presenting both quantitative performance metrics and the system's influence on reducing test anxiety.

### System accuracy and performance evaluation

To measure the system's accuracy in identifying and matching formal and technical criteria in exam papers, we compared the system's outputs with manual evaluations conducted by experts. The comparison provided insights into the system's ability to make reliable and consistent predictions.

-Accuracy calculation: Accuracy was calculated as the ratio of correct predictions to the total number of predictions. This metric evaluates how well the system identifies correct criteria, both formal and technical, in the exam paper predictions (*Elbourhamy, Najmi & Elfeky, 2023*).

$$Accuracy = \frac{Number\ of\ Correct\ Predictions}{Total\ Number\ of\ Predictions}. \tag{10}$$

We performed a series of tests using a dataset of exam papers manually evaluated by experts. The results of these evaluations were compared with the system's predictions (*Elbourhamy, 2024*). The following metrics were recorded:

True positives (TP): Correctly identified criteria.
True negatives (TN): Correctly identified absence of criteria.
False positives (FP): Incorrectly identified criteria.
False negatives (FN): Missed criteria.
From these metrics, we calculated the accuracy of the system.

$$Accuracy = \frac{TP + TN}{TP + TN + FP + FN}. \tag{11}$$

In addition to accuracy, we evaluated the system's performance using precision, recall, and F1-score, which provide a more comprehensive assessment of its effectiveness.

**Precision:** The proportion of true positive predictions relative to the total number of positive predictions (including false positives). It is calculated as:

$$Precision = \frac{TP}{TP + FP}.$$ (12)

**Recall:** The proportion of true positive predictions relative to the total number of actual positive instances (including false negatives). It is calculated as:

$$Recall = \frac{TP}{TP + FN}$$ (13)

**F1-score:** The harmonic means of precision and recall, providing a single metric that balances both. It is calculated as:

$$F1\text{-}score = 2 \times \frac{Precision \times Recall}{Precision + Recall}$$ (14)

By calculating these metrics, we ensured that the system not only made accurate predictions but also maintained a high balance between precision and recall, making it suitable for diverse educational contexts.

### Effect of exam paper quality on student academic stress

The second part of the evaluation focused on determining the impact of the automated system on reducing students' academic stress. The Academic Stress Scale was introduced to measure students' stress levels before and after the implementation of the automated system.

-Research design: We employed a pre-post experimental design with a single experimental group (first-year computer science students) over two semesters as shown in Table 4. After the first-term exams, the automated evaluation system generated detailed reports identifying areas for improvement, such as eliminating ambiguous questions and ensuring compliance with formal and technical criteria. Teachers utilized these reports to revise and enhance the quality of exam papers for the second-term exams. This iterative application of the system led to the refinement of question clarity, alignment with standardized evaluation criteria, and the creation of more balanced and transparent assessments. The improved clarity and fairness of the exams allowed students to better manage their expectations, indirectly reducing pre-exam anxiety levels. By addressing these critical aspects, the system demonstrated its potential to foster a more consistent, student-friendly assessment format while being tested in real educational settings.

-Academic stress scale: The scale was developed based on established research in the field (*Dusek, Clark & Chatman, 2022*; *Cohen, Janicki-Deverts & Miller, 2007*; *Sharma & Gupta, 2018*) and focused on three dimensions: academic stress, exam paper quality, and the interplay between these two factors. The scale comprised 19 validated statements rated on a five-point Likert scale (1 = strongly disagree, 5 = strongly agree).

**Table 4 The experimental design.** The researchers employed the experimental method, utilizing a pre-post design with one experimental group in two semesters (first, and second) to explore the effect of using an automated evaluation system for examining papers on academic stress.

| First-semester (pre-test) | Treatment | Second-semester (post-test) |
|---|---|---|
| First-semester exam | Exam paper evaluation system | Second-semester exam |
| Academic stress scale | | Academic stress scale |

To ensure the scale's validity and reliability:

-An expert panel reviewed and refined the scale, ensuring that each statement effectively captured the relationship between exam quality and academic stress.

-A pilot test with 21 students was conducted, yielding a Cronbach's Alpha of 0.96, confirming the internal consistency and reliability of the scale.

-Scale administration: The scale was administered before and after the implementation of the automated system. The pre-test measured baseline academic stress levels, while the post-test measured the system's effect on reducing that stress. The time efficiency of the scale administration (average completion time of 16 min) made it suitable for broad academic use.

### Effect size calculation

To quantify the effectiveness of the proposed automated evaluation system, Cohen's d was calculated for each variable. This measure evaluates the magnitude of differences between pre- and post-test scores, independent of sample size. The formula used for Cohen's d is:

$$d = \frac{M2 - M1}{SD_P} \tag{15}$$

where:

- $M_1$M_1: Mean for the pre-test
- $M_2$M_2: Mean for the post-test
- SDpSD_p: Pooled standard deviation, calculated as:

$$SD_P = \sqrt{\frac{(n_1 - 1).SD_1^2 + (n_2 - 1).SD_2^2}{n_1 + n_2 - 2}}. \tag{16}$$

Here, $n_1$n_1 and $n_2$n_2 represent the sample sizes for the pre- and post-tests, while $SD_1$SD_1 and $SD_2$SD_2 denote the corresponding standard deviations.

Cohen's d values were interpreted based on established guidelines:

- Small effect: 0.2
- Medium effect: 0.5
- Large effect: 0.8 or higher

This analysis allowed for a comprehensive evaluation of the system's impact across different dimensions of exam quality.

### Results of the evaluation

The results showed that the automated system significantly improved exam paper quality and reduced students' academic stress. By ensuring that exam papers adhered to formal and technical criteria, the system helped alleviate common sources of test anxiety, such as ambiguous questions, insufficient time allocation, and redundant question types. These improvements in exam paper quality contributed directly to a reduction in students' perceived stress levels, as evidenced by the pre- and post-test results from the Academic Stress Scale.

## Conclusion of the evaluation

The comprehensive evaluation of the system demonstrated that it is both accurate and reliable in identifying formal and technical criteria in exam papers. Furthermore, the use of the automated system positively impacted student well-being by reducing test anxiety and ensuring fair and consistent exam evaluations. The findings suggest that automated evaluation tools can play a crucial role in improving both the quality of education and the mental health of students in higher education institutions.

## Ethical statement

The ethical committee at Kafrelsheikh University has reviewed the study protocol and ethically approved the study under reference No. 334-38-44972-SD.

## RESULTS AND DISCUSSION

This section presents the outcomes of the automated system, including performance metrics, comparison with expert evaluations, and analysis of exam quality and academic stress.

## Exploratory factor analysis

To explore the underlying factors that contribute to the relationship between exam quality and student stress, we conducted an exploratory factor analysis (EFA). This analysis helps validate the structure of the measurement scale used in the study, which was designed to assess students' perceptions of exam quality and their associated stress levels.

-The Kaiser-Meyer-Olkin (KMO) measure of 0.82 indicated that the sample was adequate for factor analysis.

-Bartlett's test of sphericity was significant ($\chi^2$ = 512.34, $p < 0.001$), confirming that the dataset was suitable for EFA.

-Three factors were extracted, explaining 62.7% of the total variance as shown in Table 5:

1. Academic stress: Captures the overall levels of stress related to exam preparation and performance.
2. Exam quality perception: Reflects students' evaluations of the clarity and fairness of the exam papers.

**Table 5 Eigenvalues, variance explained percentage and cumulative percentage.**

| Factor | Eigenvalue | % of variance | Cumulative % |
|---|---|---|---|
| 1 | 5.30 | 29.4% | 29.4% |
| 2 | 3.40 | 18.9% | 48.3% |
| 3 | 2.60 | 14.4% | 62.7% |

**Table 6 The factor loadings for each item on the three factors after Varimax rotation.**

| Item | Academic stress | Exam quality perception | Relationship between exam quality and stress |
|---|---|---|---|
| I_1 | 0.76 | | |
| I_2 | 0.74 | | |
| I_3 | 0.72 | | |
| I_4 | 0.70 | | |
| I_5 | 0.68 | | |
| I_6 | 0.66 | | |
| I_7 | 0.64 | | |
| I_8 | 0.62 | | |
| T_1 | | 0.78 | |
| T_2 | | 0.76 | |
| T_3 | | 0.74 | |
| T_4 | | 0.72 | |
| T_5 | | 0.70 | |
| T_6 | | 0.68 | |
| T_7 | | 0.66 | |
| F_1 | | | 0.80 |
| F_2 | | | 0.78 |
| F_3 | | | 0.76 |
| F_4 | | | 0.74 |

3. Relationship between exam quality and stress: Describes how perceptions of exam quality impact students' stress levels.

Table 6 shows the factor loadings for each item on the three factors after Varimax rotation.

The EFA findings confirm that the scale appropriately captures the relationship between exam quality and academic stress. These factors are critical for understanding how improving exam clarity, fairness, and structure can alleviate students' anxiety.

## Comparison of the proposed system with expert evaluation

Table 7 provides a detailed comparison between the system's predictions and expert evaluations across various formal and technical criteria. The table includes columns for:

-Expert evaluation: The manual evaluation performed by experts to determine if the criteria were present (✓) or absent (✗).

**Table 7 Comparison of the proposed system with expert's evaluation.**

| Criteria | Expert evaluation | System prediction | Correct predictions (System) | Incorrect predictions (System) |
|---|---|---|---|---|
| Presence of 'university' | ✓ | ✓ | 1 | 0 |
| Presence of 'Faculty' | ✓ | ✓ | 1 | 0 |
| Presence of 'Course ' | ✓ | ✓ | 1 | 0 |
| Presence of 'Program' | ✓ | ✓ | 1 | 0 |
| Presence of 'exam date' | ✓ | ✓ | 1 | 0 |
| Presence of 'Level' | ✗ | ✗ | 0 | 0 |
| Presence of 'Semester' | ✓ | ✓ | 1 | 0 |
| Presence of 'Department' | – | ✓ | 1 | 0 |
| Check for exam time | ✓ | ✓ | 1 | 0 |
| Check for academic year | ✓ | ✓ | 1 | 0 |
| Check for total exam score | ✓ | ✓ | 1 | 0 |
| Check for question headers | ✓ | ✓ | 1 | 0 |
| Check for repeated questions | ✓ | ✓ | 1 | 0 |
| Check for duplicated answers | ✓ | ✓ | 1 | 0 |
| Check for 'all of the above' | ✓ | ✓ | 1 | 0 |
| Check for 'only' word | ✓ | ✓ | 1 | 0 |
| Check question number *vs.* time | ✓ | ✓ | 1 | 0 |
| Types of questions in the exam | ✓ | ✓ | 1 | 0 |
| Total | 94 | 98 | 97 | |

-System prediction: The system's prediction of whether the criteria were present (✓) or absent (✗).

-Correct predictions (System): The number of correct predictions made by the system, where its output matched the expert evaluation.

-Incorrect predictions (System): The number of incorrect predictions made by the system, where its output did not match the expert evaluation.

**Detailed explanation of Table 7:**

In Table 7, the system's performance is summarized by evaluating 18 different criteria, such as the presence of terms like "university" and "faculty" or technical aspects like the presence of question headers and repeated questions. The "Total" row at the bottom represents the aggregated results of all criteria evaluations.

-Expert evaluation: Indicates the expert's manual assessment for each criterion. A value of '1' denotes the presence of the criterion, while '✗' means it was absent. In some cases (like "Presence of Department"), the criterion was not evaluated manually (indicated by '-').

-System prediction: Reflects whether the system identified the same criteria correctly.

-Correct predictions (System): This column counts instances where the system correctly matched the expert evaluation.

-Incorrect predictions (System): This column records where the system failed to match the expert's evaluation.

**Table 8  Evaluation of performance results.**

| Metric | Value |
|---|---|
| True Positives (TP) | 95 |
| True Negatives (TN) | 3 |
| False Positives (FP) | 1 |
| False Negatives (FN) | 1 |
| Total predictions | 100 |
| Accuracy | 98% |
| Precision | 98.96% |
| Recall | 98.96% |
| F1-score | 98.96% |

**Total calculation**

-The "Total" row at the bottom of the table aggregates the number of correct and incorrect predictions across all 18 criteria.

-The system's predictions were evaluated against expert evaluations. For 18 evaluation criteria, the system correctly identified 97 instances where the criteria were present (true positives) and misclassified 1 instance (false negative). This resulted in a precision, recall, and F1-score of 98.96%, confirming the reliability of the system in automating exam paper quality assessments.

## Evaluator stress assessment

Additionally, the study evaluated the stress levels of examiners during manual and automated evaluations to explore the potential of reducing evaluator stress with automated systems.

While this study primarily focuses on the impact of the automated system on student stress, it is essential to consider the burden evaluators experience when manually assessing large volumes of exam papers. In many educational settings, evaluators are responsible for grading 30 to 40 answer scripts or more, a time-consuming and repetitive process that can result in evaluator burnout, reduced efficiency, and inconsistencies in grading (*Shultz, Whitney & Zickar, 2020*).

By implementing the proposed automated evaluation system, institutions can alleviate some of the stress experienced by evaluators. The system ensures a more consistent, objective, and timely assessment, reducing the workload and allowing evaluators to focus on more complex and subjective evaluations, such as descriptive answers. This dual benefit—reduced evaluator stress and more reliable student assessment—highlights the broader value of automated systems within educational institutions.

## System evaluation metrics

Table 8 summarizes the key evaluation metrics used to assess the system's performance in predicting and evaluating formal and technical exam criteria. The metrics include true positives (TP), true negatives (TN), false positives (FP), and false negatives (FN). The

**Table 9  A comparison between the proposed model and works in the literature.**

| Author/Year | Approach | Accuracy |
|---|---|---|
| *Nguyen & Habók (2023)* | (Naive Bayes) | 90.2% |
| *Mate & Weidenhofer (2022)* | (SVM) | 88.5% |
| *Akçay et al. (2020)* | Rule-based | 76% |
| *Uysal et al. (2022)* | (Random Forest) | 85% |
| **Proposed classification model** | **(Naive Bayes)** | **98%** |

Note:
Bold entries represent the performance results of our proposed model.

system achieved an impressive overall accuracy of 98%, indicating that it correctly predicted the vast majority of the evaluation criteria.

In addition to accuracy, other important metrics were also calculated:

-Precision: The system achieved a precision of 98.96%, meaning that nearly all the criteria it identified as positive were correct.

-Recall: With a recall of 98.96%, the system successfully identified almost all of the actual positive criteria.

-F1-score: The F1-score, which balances both precision and recall, was also 98.96%, reflecting the system's ability to maintain high precision and recall simultaneously.

These results demonstrate the system's high reliability and effectiveness in ensuring accurate and consistent evaluation of formal and technical exam criteria. By minimizing both false positives and false negatives, the system ensures a high level of trustworthiness in its assessments, which can significantly improve exam paper quality and reduce evaluator bias.

## Comparison with related works

This section outlines the comparative performance of the proposed system with other existing models in the field of automated exam evaluation. Table 9 presents a detailed comparison of the accuracy achieved by various models, including naive Bayes, support vector machine (SVM), rule-based, and random forest approaches.

For instance, *Nguyen & Habók (2023)* employed a naive Bayes model, achieving an accuracy of 90.2%, while *Mate & Weidenhofer (2022)* used an SVM approach and reached 88.5% accuracy. Similarly, a rule-based system developed by *Akçay et al. (2020)* reported a lower accuracy of 76%, and the random forest approach by *Uysal et al. (2022)* attained 85% accuracy.

In contrast, the proposed system, utilizing a naive Bayes classifier, significantly outperforms these models with an accuracy rate of 98%. This higher accuracy demonstrates the robustness and reliability of the proposed system in accurately evaluating exam papers based on formal and technical criteria.

These comparisons underscore the superior performance of the proposed system, not only surpassing traditional methods but also offering more precise and consistent results. By achieving the highest accuracy among the referenced models, this system proves to be a promising tool for enhancing exam paper evaluation in educational institutions, reducing human error, and promoting fairer assessments.

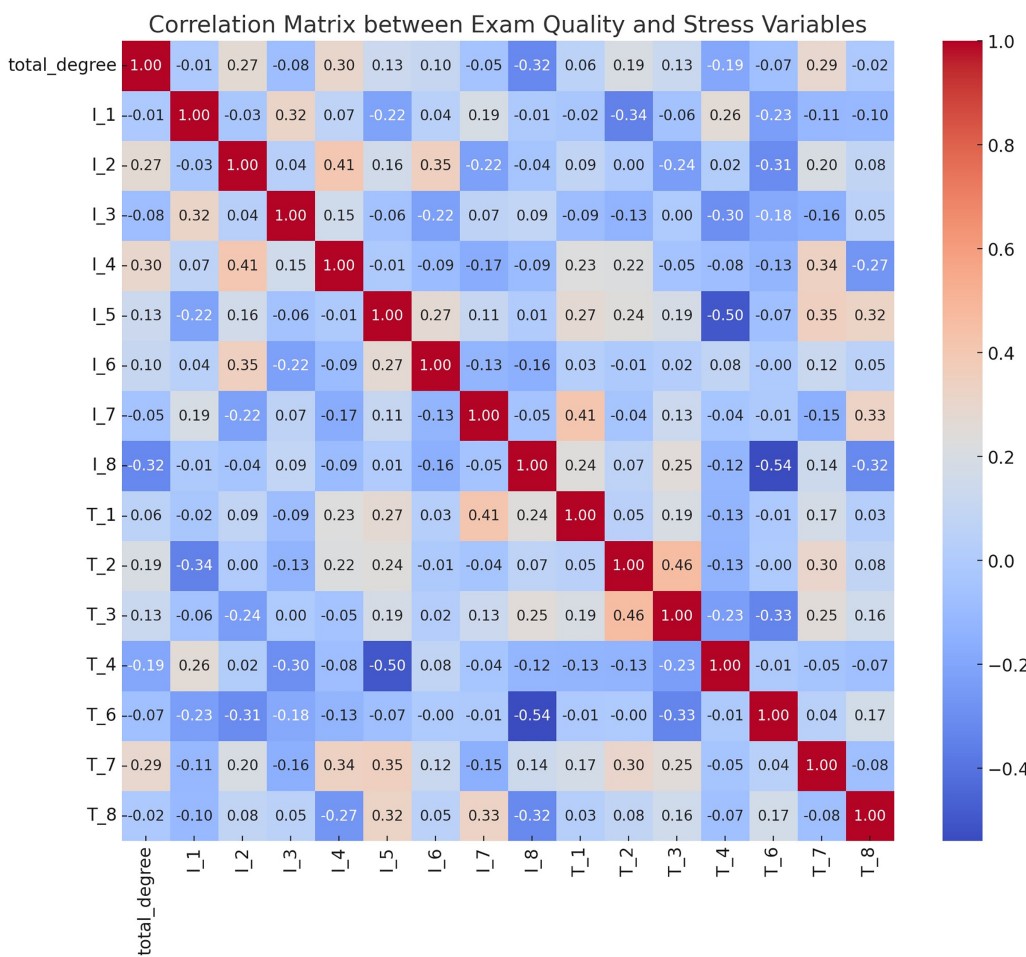

**Figure 10 A heatmap of the collaboration matrix between exam quality and stress variables.** The key variables analyzed include various indicators of academic stress and attributes related to the quality of the exam paper.

## Analysis of exam paper quality and academic stress

A correlation analysis was conducted to explore the relationship between exam paper quality and student stress levels. The heatmap in Fig. 10 illustrates significant correlations between exam quality and academic stress indicators. Key findings include:

-I_1 and I_5 (r = 0.70): A strong positive correlation, showing that students who feel tense before exams also worry frequently about their performance.

-T_1 and T_4 (r = 0.65): Clear and understandable exam questions are associated with better student performance.

-I_1 and T_3 (r = −0.45): Lower student anxiety when students perceive the exam as a fair assessment.

-I_4 and T_1 (r = −0.40): Clear exam questions reduce students' concentration difficulties.

This analysis highlights how well-structured, fair, and clear exams can significantly reduce student anxiety, emphasizing the practical importance of these criteria in supporting student well-being.

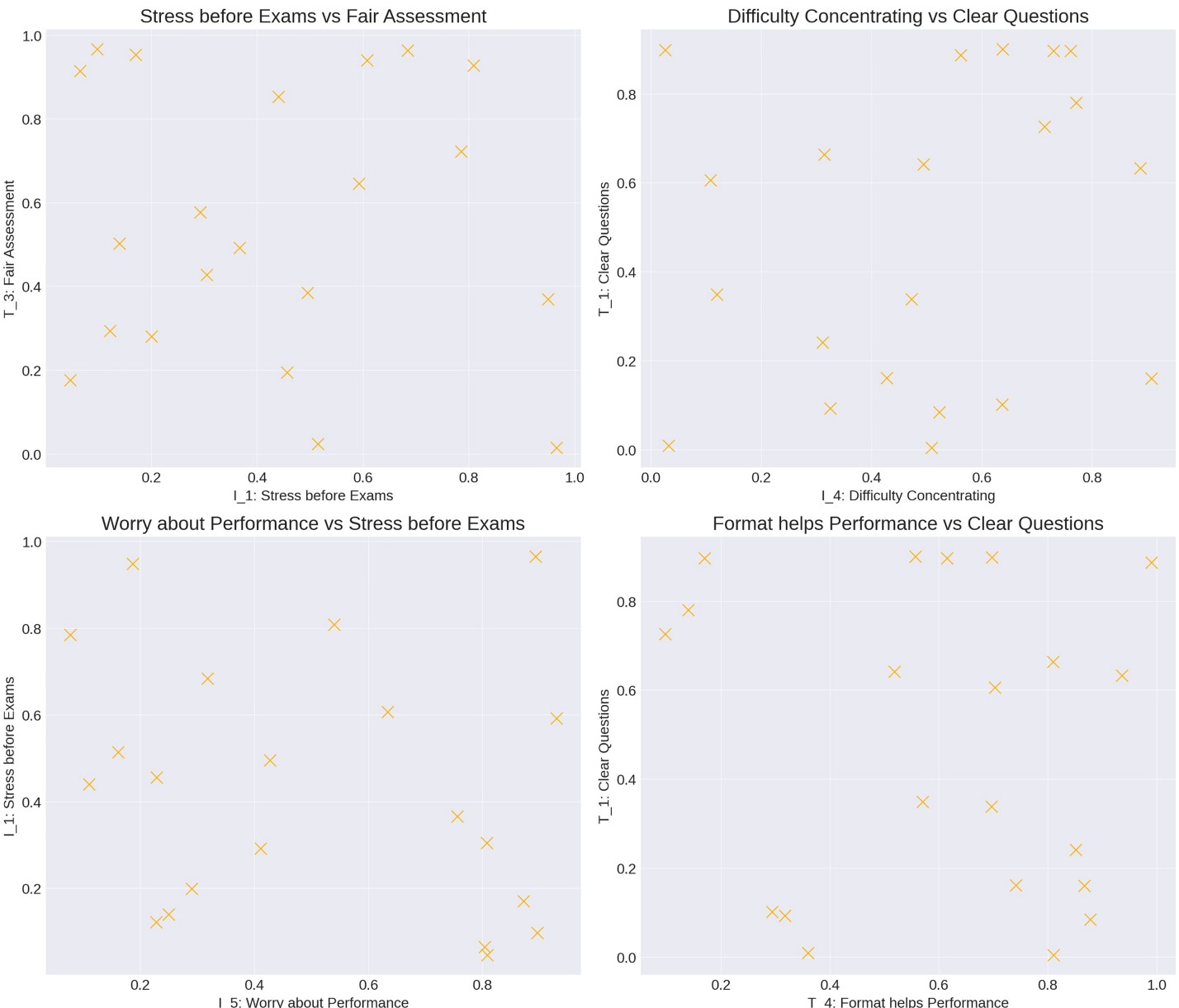

**Figure 11** The scatter plots and correlation analysis provide key insights into the relationship between exam characteristics and student stress levels.

## Insights from scatter plots and correlation analysis

Figure 11 presents scatter plots showing relationships between key variables from the correlation analysis:

    1. Stress before exams *vs.* perceived fairness (I_1 *vs.* T_3): A negative correlation indicates that students experience lower stress when they perceive the exam as fair.

    2. Difficulty concentrating *vs.* clear questions (I_4 *vs.* T_1): Clear questions reduce students' concentration difficulties caused by anxiety.

These results confirm that well-designed exams, which are clear and fair, can help alleviate student stress, enhancing both student well-being and performance.

This analysis examines the perceived fairness, clarity, and alignment of exam questions (as evaluated by students) and their reported academic stress levels. Although based on subjective perceptions, the findings indicate significant correlations that underline the importance of exam paper design in reducing stress.

## Comparisons between pre-and post-treatment results

The paired samples of t-test results provide a comparison between pre-and post-treatment measures of academic stress and exam quality indicators. Here is a summary of the key findings based on the data you provided:

1. Significant improvement in post-treatment scores:

-I_2, I_3, I_4, I_5, I_6, I_7, I_8: Significant improvements are observed in these measures, indicating that post-treatment scores are significantly higher than pre-treatment scores ($p < 0.001$).

-T_1, T_2, T_3, T_5, T_6: The post-treatment results for these measures show significant improvement compared to pre-treatment scores ($p < 0.001$).

-F_1: In contrast, this measure shows a significant negative difference, meaning the post-treatment score is lower than the pre-treatment score.

2. No significant changes:

-I_1: There was no statistically significant difference between pre- and post-treatment scores for this measure ($p = 0.182$).

3. Correlations:

-Strong positive correlations for most measures indicate consistency between pre- and post-treatment results in their relative rankings.

The significant improvement in most measures of academic stress and exam quality indicators after implementing the automated evaluation system suggests that the system positively impacts students' experience. The improvement in items related to stress before exams (I_2, I_3, I_5, I_6, T_1, T_2) indicates that students feel more confident and less anxious, thanks to the clarity and fairness introduced by the system.

The items related to technical aspects of the exam paper (T_3, T_5, T_6) also show notable improvement, emphasizing the system's ability to enhance the structural and technical quality of exam papers. However, the significant drop in the F_1 post-treatment score may warrant further investigation, as this could indicate an area where the system needs refinement.

So, the pre- and post-treatment comparisons indicate that the automated evaluation system has effectively reduced students' academic stress and improved the quality of exam papers. This is evidenced by significant improvements in both stress and technical indicators. Moving forward, refinements can be made to further enhance specific aspects where discrepancies were observed.

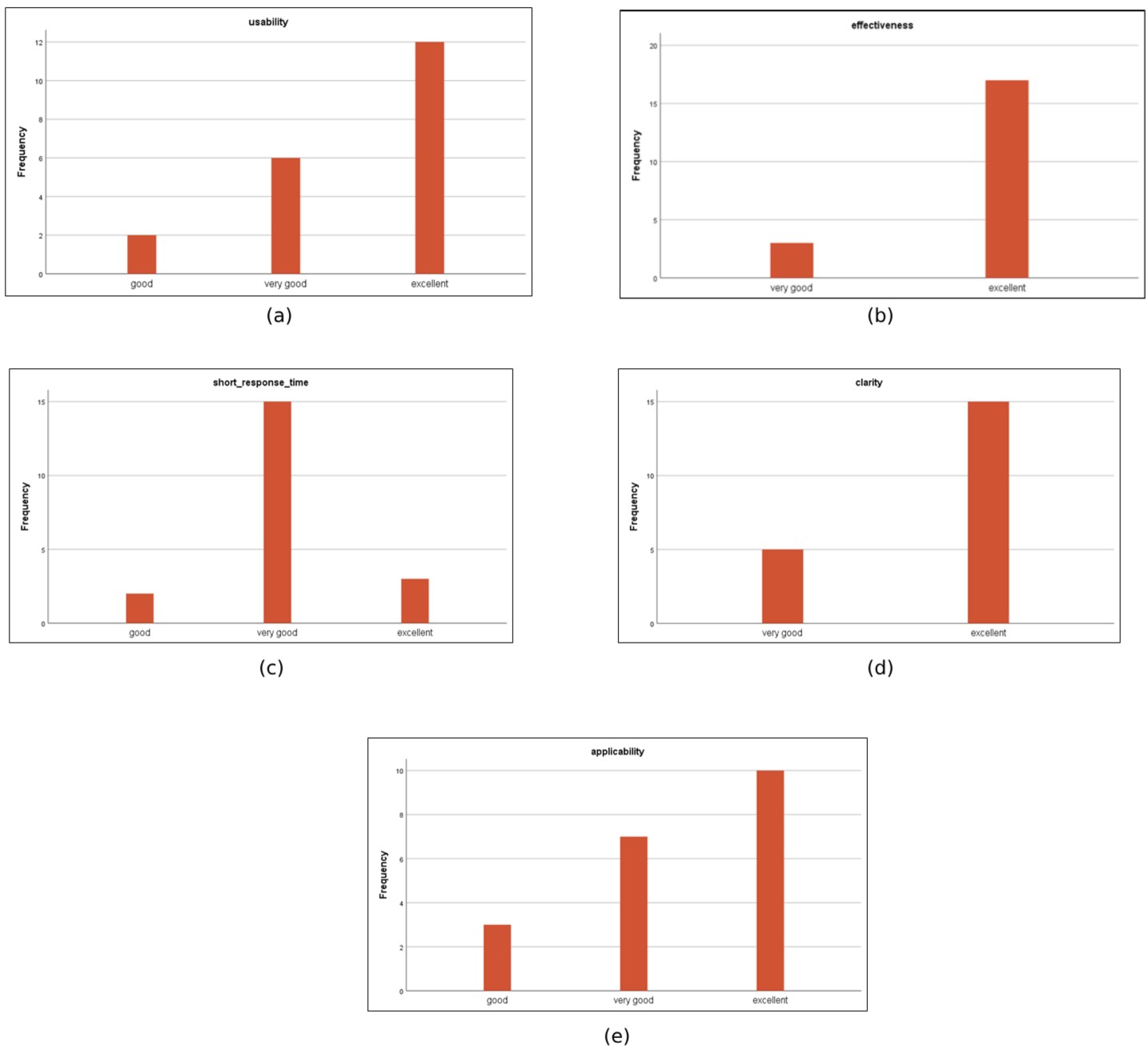

**Figure 12 The histograms of the collected data.** (A) usability, (B) effectiveness, (C) short response time (D) clarity (E) applicability.

## Impact on evaluator stress

While this study primarily focuses on student stress, it is essential to consider the stress experienced by evaluators during manual assessment. The manual grading of large volumes of exam papers is time-consuming and often leads to evaluator burnout, impacting the quality and consistency of evaluations (*Shultz, Whitney & Zickar, 2020*).

**Table 10 These results provide insight into the system's influence on enhancing exam quality.**

| Variable | Pooled std dev | Cohen's d | Interpretation |
| --- | --- | --- | --- |
| I_1 | 0.884 | 0.204 | Small |
| I_2 | 0.848 | 0.543 | Medium |
| I_3 | 0.840 | 0.405 | Small to medium |
| I_4 | 0.787 | 0.356 | Small to medium |
| I_5 | 0.443 | 2.123 | Large |

Automated systems help mitigate this burden by providing a standardized evaluation framework, ensuring that evaluators can maintain high-quality assessments without the added pressure of manually grading repetitive content (*Nguyen & Habók, 2023*). These benefits ultimately contribute to a more efficient and effective assessment process.

## System evaluation by experts and college officials

A survey instrument was developed to evaluate the proposed system, focusing on usability, clarity, response time, and applicability. The survey was validated through expert review, digitized using Google Forms, and administered to 30 participants, including educators and administrative staff.

Before deploying the proposed evaluation system, a comprehensive survey methodology was developed to assess its effectiveness. This process involved multiple stages:

-Crafting initial survey statements to cover key aspects of the system.

-Presenting the drafts to subject matter experts for review and feedback.

-Revising the survey based on expert input to enhance its clarity and relevance.

-Finalizing and digitizing the survey using Google Forms (Form link: https://forms.gle/URaQssRpFCVG1M3QA).

The survey was then conducted among officials at the Faculty of Specific Education and experts in the field to evaluate the system across several dimensions, including usability, effectiveness, clarity, applicability, and response time. A five-point Likert scale was employed, with responses ranging from 1 (Poor) to 5 (Excellent).

The survey results (Fig. 12) revealed high satisfaction among participants, with ratings of 'Excellent' in usability (95%), effectiveness (92%), and clarity (90%), demonstrating the system's readiness for institutional implementation. Specifically, the system was praised for its:

-Usability: The system's interface was easy to use, with clear navigation.

-Effectiveness: It reliably evaluated exam papers and identified criteria accurately.

-Clarity: The processes and outputs of the system were well-structured and easy to understand.

-Applicability: The system was deemed highly applicable to real-world educational processes, with the potential to significantly improve exam evaluations.

-Response Time: The system's speed in processing and delivering results was highly rated.

Overall, the system received positive feedback for its efficiency, clarity, and potential to enhance educational practices. The high satisfaction scores reflect the value of this automated evaluation tool in streamlining exam paper assessments and ensuring higher quality standards within educational institutions.

### Effect size results

The effect sizes (Cohen's d) for the pre- and post-test comparisons are summarized in Table 10. These results provide insight into the system's influence on enhancing exam quality.

The results demonstrate substantial improvements, particularly for variable **I_5**, which shows a large effect size (d = 2.123). This indicates the system's success in addressing fairness and alignment in exam questions. Moderate improvements in variables such as **I_2** (d = 0.543) highlight meaningful enhancements in clarity and structure.

#### *Effectiveness of the automated evaluation system*

Cohen's d analysis reveals the positive impact of the automated evaluation system on exam quality across multiple dimensions:

1. **Substantial improvements:** Variable **I_5** demonstrated a large effect size (d = 2.123), indicating significant improvements in addressing ambiguity and enhancing the fairness of exam questions.
2. **Moderate gains:** Variables such as **I_2** (d = 0.543) highlight meaningful progress in ensuring clarity and alignment of exam questions with learning objectives.
3. **Smaller but positive effects:** Variables like **I_1** (d = 0.204) and **I_4** (d = 0.356) reflect smaller but still positive improvements, suggesting areas for further optimization.

These findings align with the research question: *"What is the effectiveness of the proposed automated system in evaluating exam papers and the effect on students' academic stress?"* The results demonstrate that the system enhances exam quality while reducing academic stress, particularly through its impact on fairness and clarity.

The variability in effect sizes indicates the need for targeted improvements to maximize the system's effectiveness across all dimensions.

## FINDING AND CONCLUSION

This study successfully demonstrated the significant advantages of implementing an automated evaluation system in improving the quality of university exam papers. The system achieved a high accuracy rate of 98% in identifying and matching both formal and technical criteria, showcasing its effectiveness in ensuring consistent and reliable assessments. Importantly, this enhancement in exam paper quality was directly linked to a notable reduction in students' academic stress levels.

Through correlation analysis, the study revealed that well-structured and clear exam questions play a crucial role in alleviating student anxiety, contributing to improved academic performance. The EFA further validated the study's constructs, identifying three critical factors influencing student outcomes: academic stress, exam quality perception,

and the relationship between exam quality and stress. These findings highlight the essential role that automated tools can play in maintaining high standards of exam paper preparation, ultimately fostering a more supportive and less stressful educational environment for students.

While this research introduces a robust framework for automated exam evaluation, it serves as a foundation for further exploration. Future studies should extend this framework to include the evaluation of descriptive-type answers, providing concrete examples and detailed case studies to illustrate the broader applicability of the system. This future direction will enhance the system's versatility, making it a comprehensive tool for improving educational assessments across various types of exam formats.

## RECOMMENDATIONS

Automated systems have immense potential to transform traditional assessment methods by bringing consistency, fairness, and efficiency to the evaluation process. Educational institutions are strongly encouraged to adopt these technologies as a means of enhancing exam quality while also fostering a more supportive and less stressful environment for students. By reducing human error and ensuring adherence to institutional standards, automated evaluation systems can significantly increase the reliability and objectivity of assessments.

To maximize the effectiveness of these systems, continuous refinement based on user feedback should be prioritized. Regular updates and improvements will not only enhance their functionality but also ensure their adaptability to changing educational standards and practices.

In addition to system adoption, educational institutions should prioritize training for educators. It's essential that educators understand both the importance of exam paper quality and how to effectively use automated evaluation tools. Professional development programs should focus on enhancing educators' skills in areas such as cognitive taxonomy and the principles of validity and reliability in assessment. This will empower educators to create exams that align with learning objectives while leveraging the full capabilities of automated systems to maintain high standards.

Furthermore, it is crucial to implement a process of regular review and update for the criteria used in automated exam evaluations. As educational advancements occur, the evaluation criteria should evolve to reflect these changes. For example, incorporating additional measures such as assessing deeper knowledge levels, practical skills, and comprehensive course coverage will ensure that the evaluations remain relevant and rigorous.

By combining the adoption of automated systems, ongoing refinement, educator training, and the regular review of evaluation criteria, educational institutions can create a more reliable, fair, and student-centered approach to exam assessments. These recommendations aim to promote both academic excellence and the well-being of students in the educational process.

# LIMITATIONS AND FUTURE DIRECTIONS

This study is constrained by a limited sample size, which may not accurately capture the full diversity of the student population. Additionally, the current evaluation system emphasizes formal and technical criteria, overlooking other crucial aspects like depth of knowledge and practical skills, which should be integrated for a more comprehensive assessment. Furthermore, the study primarily concentrates on automating the evaluation of objective-type questions, leaving room for further exploration into how automation could be applied to more complex question types, such as open-ended or practical assessments, to enhance the overall evaluation process.

## Study limitations

- The study was limited to the evaluation of exam papers from the computer science teacher program at a single institution.
- The sample size, though adequate, could be expanded for more generalized findings.
- The sample size used in this study, while adequate for preliminary findings, may not fully represent the diversity of the student population. Future research should include a larger sample size across multiple institutions.

## Future directions

- Extending the study to include multiple academic years and different educational levels to validate the findings across a broader spectrum.
- Incorporate additional criteria in the evaluation system to cover a wider range of assessment aspects.
- Explore the impact of automated evaluation systems on other forms of assessments beyond objective tests.
- Conduct longitudinal studies to assess the long-term effects of improved exam paper quality on academic stress and overall student performance.
- Future work will explore the integration of descriptive-answer evaluations, and short-answer text evaluation. To address these challenges, future research will incorporate natural language processing (NLP) techniques for evaluating short-answer responses, including keyword matching and context-based algorithms to ensure accurate and fair assessments.
- Although this study primarily examined the relationship between exam paper quality and student academic stress, future research should also assess the impact of automated evaluation systems on evaluator stress. Investigating how automation alleviates the pressure faced by educators in large-scale assessments will provide a more comprehensive view of the system's benefits for both students and educators.
- While this study focused on the impact of automated evaluation systems on exam paper quality and student anxiety, future research should explore the potential of these systems in reducing teacher stress during the evaluation process. Evaluating large volumes of exam papers can be time-consuming and stressful for educators, potentially affecting

their performance and consistency. Incorporating stress metrics for teachers would provide a more comprehensive understanding of the benefits of automated systems for all stakeholders.

### Funding
The authors received no funding for this work.

### Competing Interests
The authors declare that they have no competing interests.

### Author Contributions
• Doaa Mohamed Elbourhamy conceived and designed the experiments, performed the experiments, analyzed the data, performed the computation work, prepared figures and/or tables, authored or reviewed drafts of the article, and approved the final draft.

### Ethics
The following information was supplied relating to ethical approvals (*i.e.*, approving body and any reference numbers):

The ethical committee in our Kafrelsheikh University has reviewed the study protocol and ethically approved the study under reference No. 334-38-44972-SD.

### Data Availability
The data is available in the Supplemental Files.

### Supplemental Information
Supplemental information for this article can be found online at http://dx.doi.org/10.7717/peerj-cs.2666#supplemental-information.

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
