# Peer review of "Automated evaluation systems to enhance exam quality and reduce test anxiety"

_PeerJ Computer Science, doi:10.7717/peerj-cs.2666_

## Round 0.1 · original submission · Major Revisions

Dear authors,
You are advised to critically respond to all comments point by point when preparing a new version of the manuscript and while preparing for the rebuttal letter. Please address all comments/suggestions provided by reviewers, considering that they should be added to the new version of the manuscript.

Kind regards,
PCoelho

Reviewer 1 ·

Basic reporting

The authors have structured the paper systematically, making it engaging for readers. The topic is also highly relevant in the current context and carries significant technical weight.

Experimental design

The methodology adopted is appropriate; however, the sample data collected may not be sufficient. The authors could have considered including a wider range of subjects.

Validity of the findings

Evaluation systems using multiple-choice and true-or-false questions have long been established in higher education for their ease of grading and ability to assess a wide range of content quickly. However, while these formats are well-understood and widely implemented, the manuscript lacks sufficient detail on how descriptive or open-ended questions are evaluated.
The authors should explain the criteria, marking schemes, and processes used to ensure consistency and fairness in grading descriptive answers, along with examples to illustrate how different responses are assessed. This would provide a clearer understanding of the approach and enhance the manuscript's comprehensiveness.

Additional comments

Good Paper needs to be more illustrative in assessing descriptive-type answers

Cite this review as

Reviewer 2 ·

Basic reporting

1. The literature and contexts need extension. See detailed feedback below:
1.1 In the Introduction, the focus of this study should be explicitly explained at the beginning that it is in the field of higher education, rather than K-12 education or other contexts. The scope of the introduction is too broad.
1.2 Lines 31-36: The manuscript rightly identifies test anxiety as a significant component of academic stress. Upon reviewing the scale used for assessing “academic stress”, it appears more aligned with measuring aspects of test anxiety. It would enhance the clarity and focus of the paper if the title and introduction were refined to specifically address test anxiety, which seems to be the actual emphasis of the study.
2 The manuscript appropriately cites the literature references provided. However, it would be beneficial to expand the literature review to include a broader range of prior studies and provide clear definitions of all key terms.
2.1 Lines 61-62: Can you summarize what has been discussed in the research presented in Table 1 and provide your synthesized statements? Specifically, an explanation on how "paper quality" is defined within the context of your study would greatly aid in establishing a robust theoretical framework.
2.2 Table 1 appears to have inconsistencies in formatting and includes only four research studies. To enhance the literature review on test evaluation, expanding this table to include a wider array of relevant studies would provide a more comprehensive overview.
2.3 Lines 67-68: Similarly, the research presented in Table 2 would benefit from further elaboration and synthesis.
2.4 A section on prior literature about automated evaluation systems is needed to strengthen the literature review. Why automated evaluation systems could help address the research gap? What has been done previously on this topic? How is the automated evaluation system defined in this study and in prior studies?
3 Some figures are unreadable due to low resolution.
3.1 Texts in Figures 3-9 and 12 are unreadable.

Experimental design

1. The research question is not well-defined. In Line 91-92, the research question is “What is the effectiveness of the proposed automated system in evaluating exam papers and the effect on student’s academic stress?”, which suggests a dual focus that may not be fully explored within the scope of the study. This research seems to consist of two complete studies, the first study explored the development and evaluation of an automated system designed to enhance test paper quality and its accuracy in a university setting. The second study is an analysis of the relationship between students' perceptions of test paper quality and test anxiety, measured via an "academic stress" scale. However, the anticipated exploration of the automated system's impact on students' academic stress, as outlined in the research question, seems to be absent.
2. I commend the authors for their detailed presentation of the automated system's development and their efforts for building the user-friendly interface. However, the methods section could benefit from further revisions to enhance clarity and coherence, and to provide additional detail. Specific feedback on the study design and methods is provided below.
2.1 Lines 112-116: I am expecting a section here to thoroughly describe the development of the criteria for paper quality. For example, why were these formal and technical criteria included? What prior literature, rationale, or theoretical framework supports their inclusion? Why is the evaluation of these aspects of the test papers important?
2.2 Lines 128: Have you evaluated the precision of the PDF-to-text process? Are the transformed results precise enough?
2.3 Lines 168-169: Could you provide more details about the training data and how it was generated? What does the training data look like? How similar is it to the real test papers?
2.4 Lines 206-214: It appears that 3 rules have been applied for identifying question patterns, while contents of exam papers at the university level usually vary significantly across majors and departments. If the university has a uniform format guide for all course lecturers, then this pattern identification would likely be effective. If not, how can you accurately capture the question patterns for tests that typically have more sophisticated text structures?
2.5 Lines 190-278: The structure of this section is quite confusing as there are no numbered headings, and the subheadings within this section are also confusing, making it difficult to understand the structure of the automated evaluation system. Additionally, lines 126-189 have already introduced the architecture of the system and the content is consistent with Figure 1. However, it is challenging to integrate the content in Lines 190-278 into the overall architecture flow. A clearer illustration of how these elements fit together would be beneficial.
2.6 Lines 285-325: This section describes the system in detail. It would be more helpful for this detailed description to appear earlier in the manuscript to aid the audience's understanding of the system at an earlier stage.

Validity of the findings

1. The findings on the evaluation metrics for this automated system and the relationship between different items within the academic stress scale are clearly presented. While the findings require further elaboration, and the presentation of results needs to be more structured and better aligned with the methods section, which currently is not the case. More detailed feedback is listed below.
1.1 Table 5 requires more detailed illustration. It appears to be an example report sheet for one exam paper evaluated by both expert evaluators and the automated system. However, the row labeled “Total” is difficult to understand.
1.2 Why are there only 100 predictions in total? Given that the sample includes 1800 questions and each question was tested by 11 criteria, it seems there should be more predictions available. Furthermore, the data set and sample that has been used for every step of data analysis need to be clearly stated in the Methods section.
1.3 Lines 440-441: Which dataset was analyzed? The “academic stress” scale was implemented in both the first and second semesters, but it remains unclear which semester’s data was analyzed for the results report.
1.4 Lines 498-507: It is unusual that the description of EFA firstly appears in the results section. This manuscript needs restructuring to ensure consistency and coherence.
1.5 Given that this study proposed a pre-post test design (according to Table 4), it is surprising that there is no comparison between the pre- and post-treatment results. Therefore, the results do not fully address one of the study's objectives: to demonstrate how the automated evaluation system might reduce academic stress. Instead, the study primarily explores the relationship between students’ perceptions of test paper quality and their test anxiety. Therefore, the narrative surrounding the study’s objectives needs to be revisited.

Additional comments

1. Overall, the scope of this study appears overly broad relative to its content, and a more focused theoretical framework would enhance its scholarly impact.
2. A significant concern is the paper's lack of coherence in presenting its objectives, methods, and results.
3. The recommendations, limitations, and future directions sections would benefit greatly from a more narrative approach, as the current use of bullet points without introductory descriptions falls short of the standards typically expected in academic research papers.

Cite this review as

·

Basic reporting

The article is well written, the context being the examination stress on students. The authors have good literature reference to the mentioned objectives.

I fail to understand the connection between student examination stress to that of automated evaluation system. The authors are encouraged to highlight the stress the evaluators undergo during evaluation, affecting the quality of evaluation. In a scenario wherein the student number is large, the evaluator has to value a minimum of 30 to 40 answer scripts.

In such a case an automated evaluation would be of great help to the education system as a whole.

Experimental design

The research objective of Automated Evaluation System is well met. My reservations are on the short answers wherein the text passage has to be a well connected list of words. TF-IDF may highlight the occurrences of terms but the context has to be analyzed.

The authors have given a detailed implementation of parsing a pdf file which might not be necessary as it does not add any value to the paper

As mentioned the stress level of the evaluator is not assessed

Validity of the findings

The stress level of evaluator has to be highlighted.
The exam stress of the student can be assessed by the Question paper quality and different ways of formulating the questions.

---

## Round 0.2 · Major Revisions

Dear authors,
After the previous revision round, some adjustments still need to be made. As a result, I once more suggest that you thoroughly follow the instructions provided by the reviewer to answer their inquiries clearly.

You are advised to critically respond to all comments point by point when preparing a new version of the manuscript and while preparing for the rebuttal letter. All the updates should be included in the new version of the manuscript.

Kind regards,
PCoelho

Reviewer 2 ·

Basic reporting

Thank you for your thoughtful efforts in addressing many of the previous comments. This manuscript has improved, particularly in illustrating the system architecture and workflow, showcasing a valuable contribution to automated evaluation tools. However, some prior comments remain insufficiently addressed. Below are more detailed comments based on the revised manuscript.

The Introduction would still benefit from additional clarification and support from relevant literature to strengthen its foundation.
1. Please include references to support these statements, “The quality of exam papers plays .... Poorly designed exams ....” (Lines 38-40) “These systems provide consistent, objective evaluations....” (Lines 44-46).
2. Due to the lack of sufficient references and theory support, I’m not convinced of this statement “automated systems can directly contribute to reducing student test anxiety” (lines 46-47). The authors also noted that “the primary aim of the system is to support teachers in evaluating exam papers efficiently.” (lines 277-278) However, it is unclear how this process directly impacts student anxiety. Did teachers modify the exam papers multiple times based on the system’s reports in this study? If yes, more detail should be included in the Experimental Design section. If not, how would this directly alleviate students' anxiety?

The Related Work still needs expansion.
3. I appreciate the expanded Table 1 and additional discussion on related work; however, the literature review could still be improved by synthesizing prior findings. For instance, while the authors highlight a gap in research on exam quality’s impact on student well-being, further explanation of the theoretical basis—such as cognitive or educational theories linking test quality to anxiety—would better contextualize this research.
4. Automated Evaluation Systems (Lines 93 and 121). A more explicit definition of "automated evaluation systems" is needed. Notably, differentiating this system’s function (e.g., text-based exam content analysis for exam quality assessment) from typical AI-based scoring engines of students’ test results is critical. It would also be beneficial to clarify intended system users (faculty, administrators, or students) early in the introduction.

Experimental design

5. System Evaluation and Real-World Application (Section 3.3.2). It is unclear whether teachers implemented changes on their originally designed exam papers based on feedback from the automated evaluation system (see Point 2).
6. Line 209 (See prior comment 2.3). Details about the training data used for model development are not included in this revision.
7. A methods section that clearly presents all analyses in the correct order is missing, which makes the overall analytical flow difficult to follow. For example, Section 3.4.2 describes the Academic Stress Scale, while Section 3.4.3 directly concludes that "the automated system significantly improved exam paper quality and reduced students’ academic stress." It would be more logical to include the analytical process and results reporting in between these sections to enhance clarity and flow. Additionally, the EFA results (Line 498) require a corresponding method description to provide context.
8. Overall, the findings of this study do not fully align with the revised research question: “What is the effectiveness of the proposed automated system in evaluating exam papers and the effect on students’ academic stress?” (Lines 91-92). There is no examination of effect sizes or the magnitude of improvements in exam paper quality or reductions in student academic stress after the implementation of this automated evaluation system.

Validity of the findings

9. Table 7. The authors added the statement, “out of 18 evaluations, the system correctly identified 97 of the criteria (Correct Predictions) and made 1 incorrect prediction (Incorrect Predictions)” (lines 465-466). However, this explanation remains unclear.
10. Correlation Analysis (Lines 515-516). The analysis seems to examine the relationship between students' perceptions on exam quality and stress levels rather than actual exam quality evaluated by the automated system. The description needs to be more accurate.
11. Evaluator Stress Assessment (Lines 468-480 and lines 566-573). The “Evaluator Stress Assessment” section appears unexpectedly in the results, without prior mention, and may not directly align with the study’s primary objectives. If included, its relevance should be established earlier in the paper.
12. System Evaluation by Experts and College Officials (lines 574-596). This section introduces new information in the results without prior mention of the methods. Describing this survey in the experimental design would enhance transparency.

Additional comments

13. Structure and Flow. The overall structure could be refined for readability. Arranging the methods and results sections in an order that reflects the experimental or analytical process would improve coherence. For instance, a methods-results pairing (e.g., "Analysis 1, Analysis 2, Analysis 3”, followed by "Results for analysis 1, analysis 2, and analysis 3", etc.) would help readers track the study’s progression.
14. Overall, this manuscript presents a valuable attempt to develop an automated evaluation system. The insights provided are meaningful; however, the paper would benefit from refinements in structure, alignment between the theoretical framework, objectives, method reporting, and results, as well as improvements in academic language. Thank you again for your diligent efforts in improving this work.

Cite this review as

·

Basic reporting

The changes made are justified.

Experimental design

The changes made are justified.

Validity of the findings

No suggestions

Additional comments

The review process has reworded the title of the paper which matches with the work done.

---

## Round 0.3 · accepted · Accept

Dear authors, we are pleased to verify that you meet the reviewer's valuable feedback to improve your research.

Thank you for considering PeerJ Computer Science and submitting your work